# Cryo-EM structures of recombinant human sodium-potassium pump determined in three different states

Yingying Guo[1,2,3,4], Yuanyuan Zhang[1,2,4], Renhong Yan [1,2,3,4], Bangdong Huang[1,2], Fangfei Ye[1,2], Liushu Wu[1,2], Ximin Chi [1,2], Yi shi[1,2] & Qiang Zhou [1,2 ✉]

Sodium-Potassium Pump ($Na^+/K^+$-ATPase, NKA) is an ion pump that generates an electrochemical gradient of sodium and potassium ions across the plasma membrane by hydrolyzing ATP. During each Post-Albers cycle, NKA exchanges three cytoplasmic sodium ions for two extracellular potassium ions through alternating changes between the E1 and E2 states. Hitherto, several steps remained unknown during the complete working cycle of NKA. Here, we report cryo-electron microscopy (cryo-EM) structures of recombinant human NKA (hNKA) in three distinct states at 2.7–3.2 Å resolution, representing the E1·3Na and E1·3Na·ATP states with cytosolic gates open and the basic E2·[2K] state, respectively. This work provides the insights into the cytoplasmic $Na^+$ entrance pathway and the mechanism of cytoplasmic gate closure coupled with ATP hydrolysis, filling crucial gaps in the structural elucidation of the Post-Albers cycle of NKA.

---

[1] Westlake Laboratory of Life Sciences and Biomedicine, Key Laboratory of Structural Biology of Zhejiang Province, School of Life Sciences, Westlake University, 18 Shilongshan Road, Hangzhou 310024 Zhejiang Province, China. [2] Institute of Biology, Westlake Institute for Advanced Study, 18 Shilongshan Road, Hangzhou 310024 Zhejiang Province, China. [3] Present address: Department of Biochemistry, School of Medicine, Southern University of Science and Technology, Shenzhen, Guangdong, China. [4] These authors contributed equally: Yingying Guo, Yuanyuan Zhang, Renhong Yan. ✉email: zhouqiang@westlake.edu.cn

The sodium potassium pump (Na⁺/K⁺-ATPase, NKA) is expressed in the plasma membrane of all animal cells and belongs to the P2-type ATPase family[1–3]. NKA converts energy derived from ATP hydrolysis into an electrochemical potential gradient of $Na^+$ and $K^+$ across the plasma membrane, which is required for numerous physiological processes[4]. NKA undergoes an ATP-driven transport cycle called the Post-Albers scheme (Supplementary Fig. 1)[5,6], during which NKA exchanges three cytoplasmic $Na^+$ for two extracellular $K^+$ through transition between the E1 and E2 states coupled with hydrolysis of one ATP molecule. The E1 state of NKA has a higher affinity for $Na^+$, while the E2 state has a higher affinity for $K^+$.

NKA consists of a large multi-transmembrane catalytic α subunit, a single transmembrane β subunit, and an auxiliary γ subunit called FXYD. There are four, three and seven isoforms for the α, β and γ subunits of human NKA (hNKA), respectively. The α subunit contains 10 transmembrane helices (M1–M10) and three cytosolic domains: the actuator (A) domain, the nucleotide-binding (N) domain, and the phosphorylation (P) domain. The A, N, and P domains of the α subunit form the cytoplasmic headpiece, which undergoes cycles of phosphorylation and dephosphorylation, resulting in closed and open conformation of the headpiece, respectively[6,7]. The reported crystal structures of pig and shark NKA in the E2P·2K and E2·Pi·[2K] states with and without cardiotonic steroids provided valuable information for understanding the extracellular cation pathway[8–12]. In contrast, only two structures in the E1 state (E1~P·[3Na]·ADP, Protein Data Bank (PDB) ID: 3WGU[13] and 4HQJ[14]) have been reported, revealing an aspartyl-phosphorylated intermediate state with three cytoplasmic $Na^+$ occluded. NKA undergoes large conformational changes during $K^+$ release and $Na^+$ access to the ion-binding sites and is occluded by cytoplasmic gate closing. Therefore, questions about the cytoplasmic $Na^+$ entrance pathway and the gating mechanism on the cytoplasmic side have remained unanswered.

In this work, we report three cryo-electron microscopy (cryo-EM) structures of NKA in different states (E1·3Na, E1·3Na·ATP and E2·[2K]) of the Post-Albers cycle, which provides the structural insights into the cytoplasmic substrate entrance and the mechanism of cytoplasmic gating coupled with ATP hydrolysis, and improves the structural elucidation of the Post-Albers cycle of NKA.

## Results

### Structural determination of hNKA in three conformations.
To solve the structures of hNKA and investigate its cytoplasmic gating mechanism, we overexpressed and purified hNKA that contained α1, β1 and γ2 subunits (Supplementary Fig. 2a). To trap hNKA in the E1 state, only $Na^+$ and $Mg^{2+}$ were added to the buffer during purification[15]. In the same way, the existence of only $K^+$ in solution can stabilize the conformation of hNKA in the E2 state[15]. We determined the cryo-EM structures of hNKA under three conditions: 1) 150 mM NaCl, 3 mM $Mg^{2+}$, 2) 150 mM NaCl, 3 mM $Mg^{2+}$, 1 mM ATP analogue adenosine 5′-O-(3-thio) triphosphate (ATPγS), and 3) 100 mM KCl, 3 mM $Mg^{2+}$ (Supplementary Figs. 2–4 and Supplementary Table 1). The cryo-EM structures solved under these conditions represent the E1·3Na, E1·3Na·ATP and E2·[2K] states at global resolution of 3.2 Å, 2.9 Å, and 2.7 Å, respectively, with higher local resolution in the transmembrane region. The map quality is good enough for model building except for some minor disordered regions. Some lipid-like densities were found near site A and site C[16], for which the cholesterol analogue CHS and phosphatidic acid 3PH was manually built according to the purification conditions (Supplementary Fig. 5).

### Overall structure of hNKA in E1 states.
We successfully solved the cryo-EM structures of hNKA in the E1·3Na and E1·3Na·ATP states preceding ATP hydrolyzed at 3.2 Å and 2.9 Å resolution in the absence or presence of a slowly hydrolyzed ATP analogue (ATPγS), respectively, which represent two E1 states of NKA preceding ATP hydrolysis previously unknown according to our knowledge (Fig. 1 and Fig. 2a, Supplementary Fig. 6a). Intriguingly, we observed an open pathway between the cytoplasm and $Na^+$ binding sites, resembling the $Ca^{2+}$ pathway in the E1·ACP·$Mg^{2+}$ (PDB ID:4H1W) state of SERCA[17]. The entrance of the open cytoplasmic cavity is located above the M1 kink and surrounded by M2, M3 and M4.

The structures of the E1·3Na and E1·3Na·ATP states are very similar to each other, with a root mean square deviation (RMSD) of 1.388 Å of 976 Cα atoms from α subunit, except for a slight difference in the $Na^+$ binding sites at the transmembrane region and in the ATP binding pocket at the cytoplasmic headpiece of the α1 subunit (Fig. 2b–d, Fig. 3a–c and Supplementary Fig. 6b, 7a, 8a–c). In the E1·3Na and E1·3Na·ATP structures, NKA has three $Na^+$ binding sites, namely, I, II and III, which are surrounded by M4, M5, M6 and M8 (Fig. 2b–d). Compared with the E1~P·[3Na]·ADP state of pig NKA, sites I and II are different, whereas site III seems very conserved in those three E1 basic states (Fig. 2e; Supplementary Fig. 8c, d). It seems that during E1·3Na, E1·3Na·ATP to E1~P·[3Na]·ADP transition, ATP binding and hydrolysis affect $Na^+$ binding at sites I and II.

The E1·3Na and E1·3Na·ATP structures present two snapshots of the open conformation of the cytosolic headpiece without direct interaction between the A and N domains, which are significantly different from the E1~P·[3Na]·ADP state. The A domain is separated from the N domain and stabilized only by the eighth helix (Pα8) at the end of the P domain. However, in the E1~P·[3Na]·ADP state, the A domain interacts with the N domain through two additional sites, stabilizing the closed conformation of the headpiece[13] (Supplementary Fig. 8e). Importantly, the movement and rotation of the A domain pull up the M1 helix after phosphorylation to close the cytoplasmic gate (Fig. 4a). However, in the Post-Albers cycle of SERCAa, a $Ca^{2+}$-ATPase, the cytosolic headpiece in the equivalent state E1·ACP·$Ca^{2+}$ (PDB:1T5S)[18] is patterned with a different closure conformation (Supplementary Fig. 12).

### Na⁺-binding sites and an open cytoplastic accessible pathway.
Since there are several nonprotein densities shown in two E1 cryo-EM maps at the mouth of the open accessible tunnel, we built the three most likely $Na^+$ locations by thorough analysis surrounding coordinating residues and comparison with the reported E1~P·[3Na]·ADP structure (Fig. 2 and Supplementary Fig. 7a Supplementary Tables 2 and 3). It seems that ion-binding sites I and III juxtaposed at M5, M6 and M8 are more reliable. Two residues, Ser782 on M5 and Asp815 on M6, are important for sites I and III formation. Site III is located between M5, M6 and M8 and the most restricted among three $Na^+$ binding sites (Fig. 2d–e, Supplementary Fig. 8a–d). The bulky side chain of Tyr778 on M5 forms a roof by cation-π interactions with site III $Na^+$(Supplementary Fig. 7a). The Thr781, Thr814 and Gln930 residues are involved in $Na^+$ coordination at site III in the E1·3Na state, whereas the Asp933 residue is close to and coordinated with $Na^+$ in the E1·3Na·ATP state. Comparison between the E1·3Na and E1·3Na·ATP states shows that the binding of ATP causes a subtle shift of $Na^+$ at site I, which are coordinated with Ala330, Ser782, Asn783, Thr779 and Asp815 in the E1·3Na state and that are coordinated with Ala330, Ser782, Asn783, Glu786, Asp811 and a water molecule in the E1·3Na·ATP state.

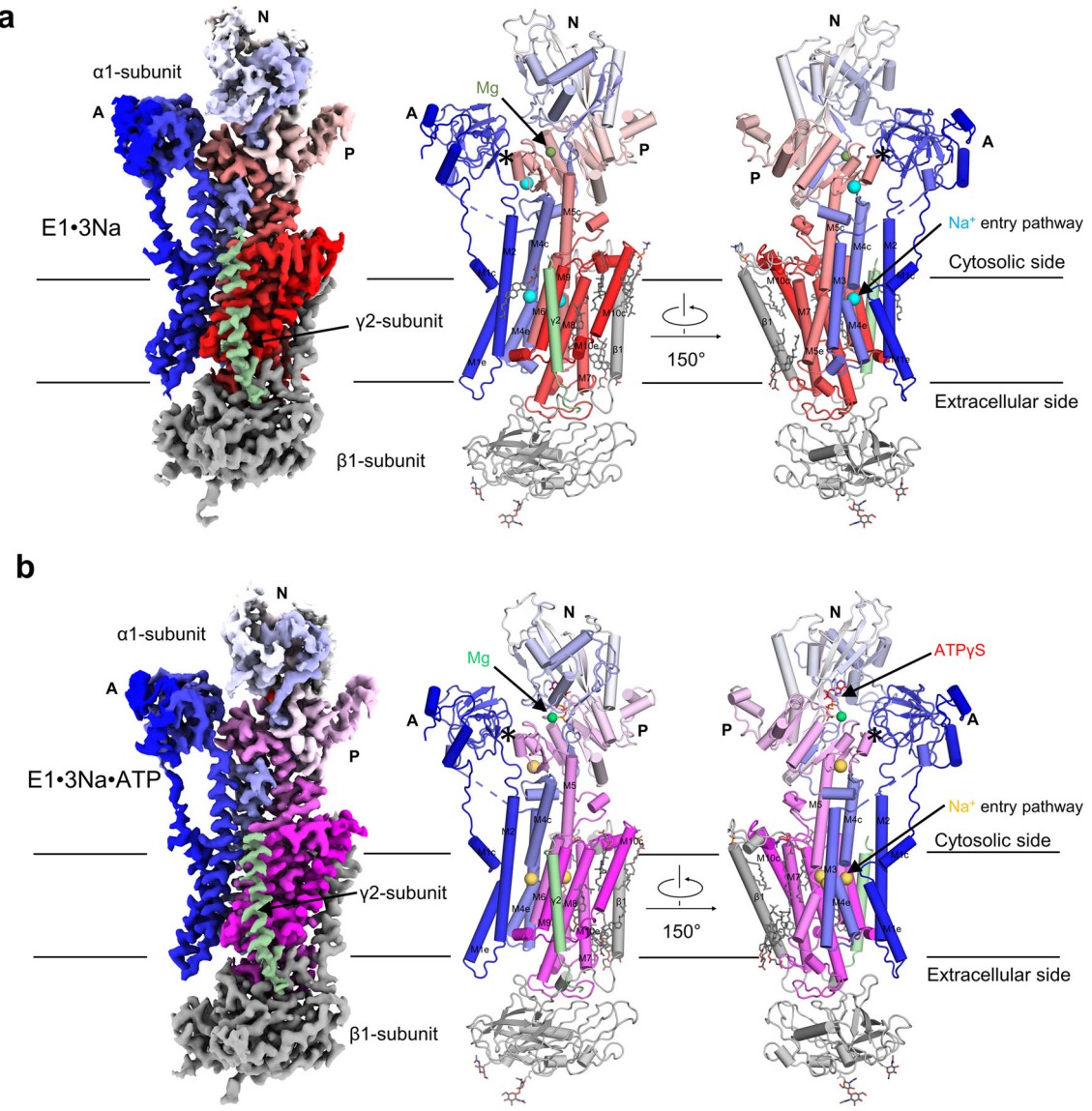

**Fig. 1 Overall structures of NKA in the E1·3Na and E1·3Na·ATP states. a** The overall cryo-EM map (left panel) and two views of the overall structure (middle and right panel) of the E1·3Na state. The α1 subunit is rainbow coloured from blue for the N-terminus to red for the C-terminus. β1 subunit is coloured grey. γ2 subunit is coloured pale green. **b** is the same as (**a**) but for the E1·3Na·ATP state (coloured as a spectrum from blue for the N-terminus to magenta for the C-terminus). Na+ are coloured cyan or yellow in (**a**) or (**b**), respectively. Mg2+ are coloured forest green or green in (**a**) or (**b**), respectively. The glycosylation moieties are shown as sticks. e extracellular, c cytosolic, M transmembrane helix.

We calculated partial valence[19] for Na+ at sites I, II, and III in the E1·3Na state, which is 0.493, 0.397, and 0.653, respectively (Supplementary Table 2), indicating site III is a reasonable coordination site for a sodium ion. The relative lower valence values for site I, II (less than to 0.5[20]) imply that sodium ion does not bind at these sites in a stable way. Whereas in the E1·3Na·ATP state, the partial valence for sites I, II, and III is 0.748, 0.313, and 0.717, respectively (Supplementary Table 3), increased at site I and III and decreased at site II.

The densities of Na+ and the side chain of the gating residue Glu334 were not well resolved at site II in either E1·3Na or E1·3Na·ATP, while the density of other residues around the entrance of the cytoplasmic cavity was well resolved (Fig. 2b–d and Supplementary Fig. 7a). A possible Na+ at site II according to surrounding coordinating residues Val329, Val332, Asp811 and waters was proposed. Glu334 has been reported as a gatekeeper that is highly conserved among P2-ATPases[18,21,22] (Supplementary Fig. 13). Although previous studies have observed that

Glu334 is a part of site II for cation binding in the closed conformation[18], we could not identify a stable conformation for the side chain of Glu334 due to weak density (Supplementary Figs. 4d, 7a). We speculate the side chain of Glu334 is unstable for the formation of Na+ coordination. Large movement of site II Na+ and the gating residue Glu334 was observed during the transition from the E1·3Na·ATP state to the E1~P·[3Na]·ADP state (Fig. 2e). Therefore, we speculate the gating residue Glu334 on M4, which has an unstable side chain, allows Na+ entrance before the M1 sliding gate closure. Structural comparison between different E1 states shows that the Phe100 and Leu104 residues of hNKA (corresponding to residues Phe93 and Leu97 in pig NKA) are away from Glu334 in the E1·3Na·ATP state, allowing Glu334 to located in the open mouth of the entrance pathway of Na+ (Supplementary Fig. 8a–d).

**The binding mode of ATPγS.** P2-Type ATPase can be trapped in diverse stages of the Post-Albers catalytic scheme using ATP

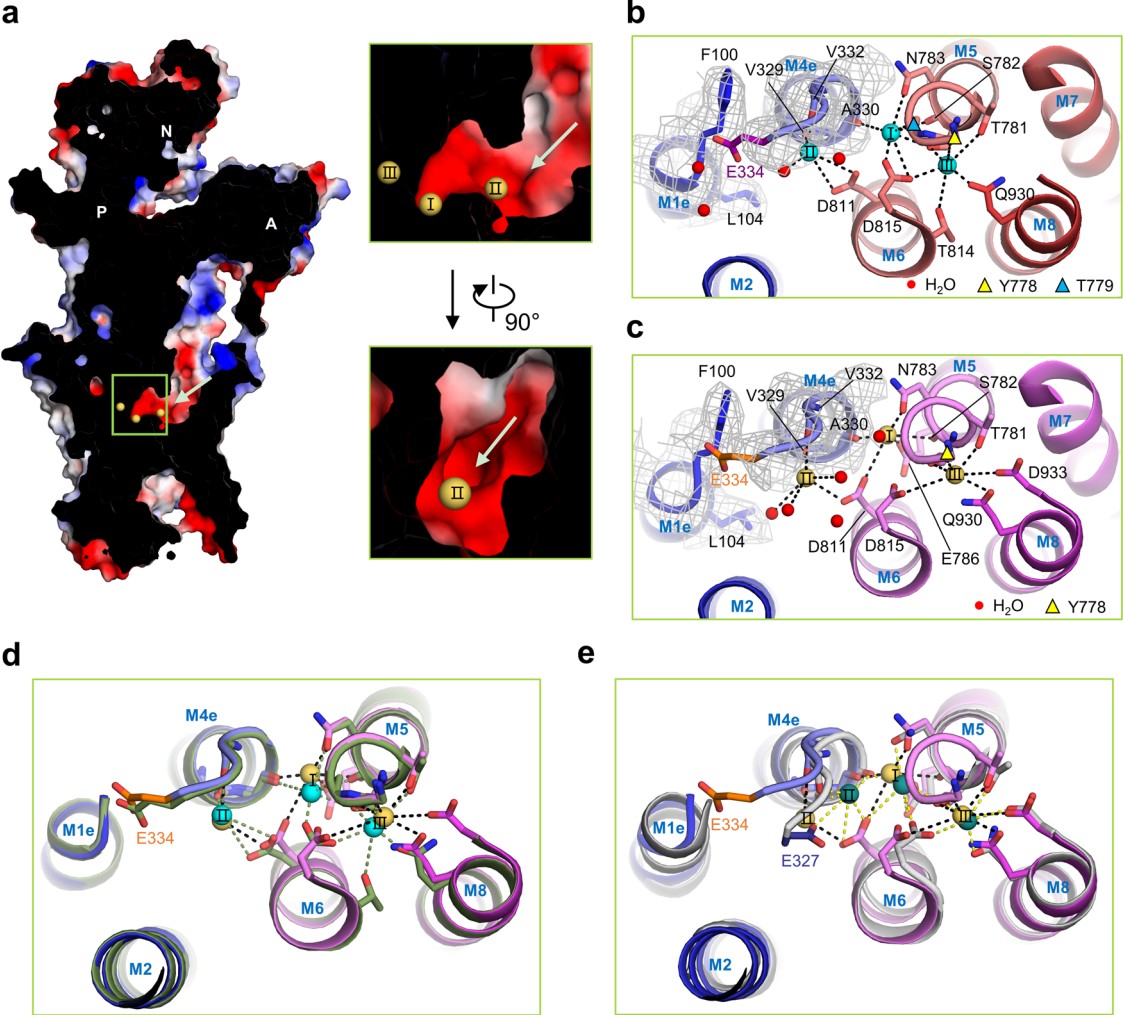

**Fig. 2 Na+ binding sites in different E1 states. a** Cross-section through the Na+ entry pathway in the E1·3Na·ATP state. Surface representation of the hNKA α1 subunit is coloured by vacuum electrostatics. Na+ are coloured yellow. **b**, **c** are the transmembrane Na+ binding sites in the E1·3Na and E1·3Na·ATP states, with Na+ (labelled I, II and III) shown in cyan and yellow, respectively. The EM density around gate residue Glu334 are shown in grey mesh. The key residues coordinated with Na+ within 4 Å are shown in sticks and indicated by dashed lines. The water molecules are shown as red balls. The yellow and blue triangles represent Tyr778 and Thr779 above the pictured field of view, respectively. **d** Binding mode comparison of Na+ between the E1·3Na (dark green) and E1·3Na·ATP (spectrum from blue to magenta) states. **e** Binding mode comparison of Na+ between the E1·3Na·ATP (spectrum from blue to magenta) and E1·P·[3Na]·ADP (grey) states. The gating residue Glu334 in human (Glu327 in pig) is coloured purple (E1·3Na), orange (E1·3Na·ATP), and slate (E1·P·[3Na]·ADP), respectively. Na+ in E1·P·[3Na]·ADP are coloured deep teal. The structures in (**d**) and (**e**) are superimposed using M7–M10.

analogues, structural analogues of phosphorylation and different ionic conditions[6,7]. In this study, ATPγS was added to hNKA to mimic the ATP-bound E1 state instead of AMPPCP (ACP)[18,23] or AMPPNP[23,24] in SERCA.

In E1·3Na·ATP, the adenine ring of the bound nucleotide is stacked with Phe482 and interacts with Lys508 and Asp450 through π-π interaction, cation-π interaction, or hydrogen bond interaction (Fig. 3a). The ribose moiety is stabilized by Asp619 and Arg692, and the phosphate moiety interacts with Arg551 and Gly618 to maintain the kinked conformation of ATPγS. The γ-phosphate of ATPγS inserts into the P domain and is stabilized by the cofactor Mg2+. The phospho-acceptor Asp376 in the highly conserved 376DKTG motif is primed to receive the γ-phosphate from ATP, while its side chain is oriented to Lys698 through a direct interaction, as in the E1·3Na state (Fig. 3b, c). Comparing ATPγS with ADP and the stable phosphate analogue aluminium fluoride, we found that they adopt a similar binding mode (Fig. 4b). However, the orientation of the Asp376 side chain is

different from the upright conformation in E1~P·[3Na]·ADP (Fig. 3d). In addition, the terminal γ-phosphate of ACP in the E1·ACP·2Ca2+ state of SERCA[18] interacts with the side chain of Asp351 (Asp376 in hNKA), which also shows an upright conformation resembling the case of E1~P·[3Na]·ADP (Supplementary Fig. 9a). The binding mode of ATPγS in the E1·3Na·ATP state of hNKA is very similar to ACP in the E2·ACP state of SERCA2a, in which the Asp376 side chain faces Lys698 (Fig. 3e). We speculate that the reason for this difference is that ATPγS binding can mimic a situation preceding NKA phosphorylation. As a result, ATPγS is delivered to the phosphorylation site with a proper orientation to facilitate phosphoryl transfer.

**Cytoplasmic gate closure coupled with ATP hydrolysis.** Both the E1·3Na and E1·3Na·ATP states of hNKA have an open cytoplasmic pathway and an open headpiece (Supplementary Fig. 8). After ATP hydrolysis, the cytoplasmic gate is closed in

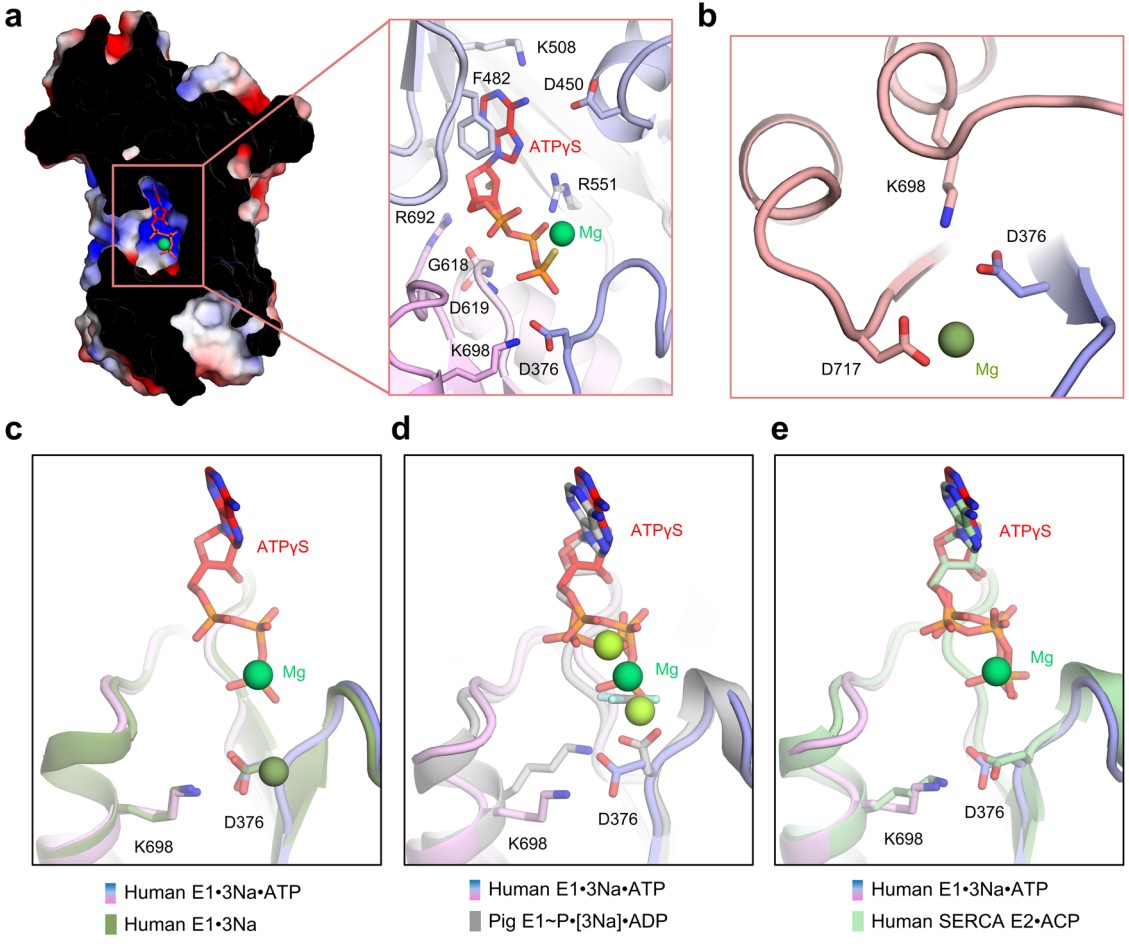

**Fig. 3 ATP binding modes in NKA. a** ATP binding pocket between the N and P domains and the key residues around ATPγS (red). **b** $Mg^{2+}$ binding site and the phosphorylation site (Asp376) in the E1·3Na state. **c** Comparison of the $Mg^{2+}$ binding site and the phosphorylation site (Asp376) between the E1·3Na (dark green) and E1·3Na·ATP (blue to magenta) states. **d, e** Comparison of the ATP binding pocket between the E1·3Na·ATP (blue to magenta) and E1~P·[3Na]·ADP (grey) or SERCA E2·ACP (pale green) states. $Mg^{2+}$ are coloured forest green, green and pea green in E1·3Na, E1·3Na·ATP and E1~P·[3Na]·ADP, respectively.

the E1~P·[3Na]·ADP state, in which the cytosolic headpiece is also closed. ATP hydrolysis takes place in the cytosolic headpiece, spatially more than 50 Å distant to the $Na^+$ binding sites at the transmembrane region. It is important to know how ATP hydrolysis is coupled with the closure of cytoplasmic gate. The P domain of NKA have a Rossmann fold core, which contains a central parallel β-sheet with seven strands (Pβ1–Pβ7) and associated α-helices (Pα2–Pα7)[25]. Comparison among three NKA E1 structures aligned to their N-terminal half of the P domain ($P_N$ domain, comprises Pβ1–4) revealed that nucleotide binding induces rotation of the N domain with respect to the P domain using Arg692 in $P_N$ domain as a pivot point, which is on a short loop connecting the third strand (Pβ3) and the sixth helix (Pα6) (Fig. 4b, Supplementary Fig. 9b, c). The ATP hydrolysis-coupled movement of the C-terminal half of the P domain ($P_C$ domain, comprises Pβ5–Pβ7) towards Asp376 changes the orientation of the Asp376 side chain (Fig. 3d) and the orientation of the A domain that sits on the Pα8 helix of the P domain (Fig. 4b, c). Finally, the movement of the N and A domains closes the cytosolic headpiece, pulls up the M1 helix after phosphorylation and closes the cytoplasmic gate (Fig. 4a).

**$K^+$-occluded E2 basic state**. In addition to the E1·3Na and E1·3Na·ATP states described above, we also solved a cryo-EM structure of hNKA in the E2·[2K] state at 2.7 Å resolution, which is

in a ground occluded state bound with two $K^+$ following dephosphorylation (Fig. 5 and Supplementary Fig. 3b). A cytoplasmic $K^+$ site (site C) is the third $K^+$ binding site (Fig. 5a), which is implicated in the activation of dephosphorylation[26]. Compared to the E2·[2K]·Pi state (a preceding state in the Post-Albers cycle)[8], the two structures are very similar, with an RMSD of 1.083 Å between 994 Cα atom pairs of α subunit (Supplementary Fig. 10). The $K^+$ binding sites I and II between M4, M5 and M6 are almost identical (Fig. 5b and Supplementary Fig. 7b, 10c). The $K^+$ coordinating residues at site I (Asp811, Thr779, Ser782, Glu786 and Asn783) and site II (Asp811, Glu334, Val332, Val329, Ala330, Glu786 and Asn783) are similar to those at $Na^+$ binding sites I and II in the E1·3Na state (Supplementary Fig. 11). The [219]TGES motif of the A domain, which plays an important role in the dephosphorylation of E2P to E2 transition[27–29], is further stabilized by $Mg^{2+}$ and [376]DKTG motifs in the P domain (Fig. 5c, d). Compared with the E1·3Na state, the α1 subunit undergoes a large conformational change (Supplementary Fig. 11c). In the TM region, M1 to M6 rotate towards the opposite side by approximately 20°, whereas M7 to M10 are relatively rigid. When the $P_N$ domain is aligned, the N domain tilts 99°, and the A domain rotates approximately 71° relative to the E1·3Na state (Supplementary Fig. 11a). The [219]TGES motif is moved to expose the conserved Asp376 phosphorylation site during the transition from the E2·[2K] to E1·3Na state (Supplementary Fig. 11c).

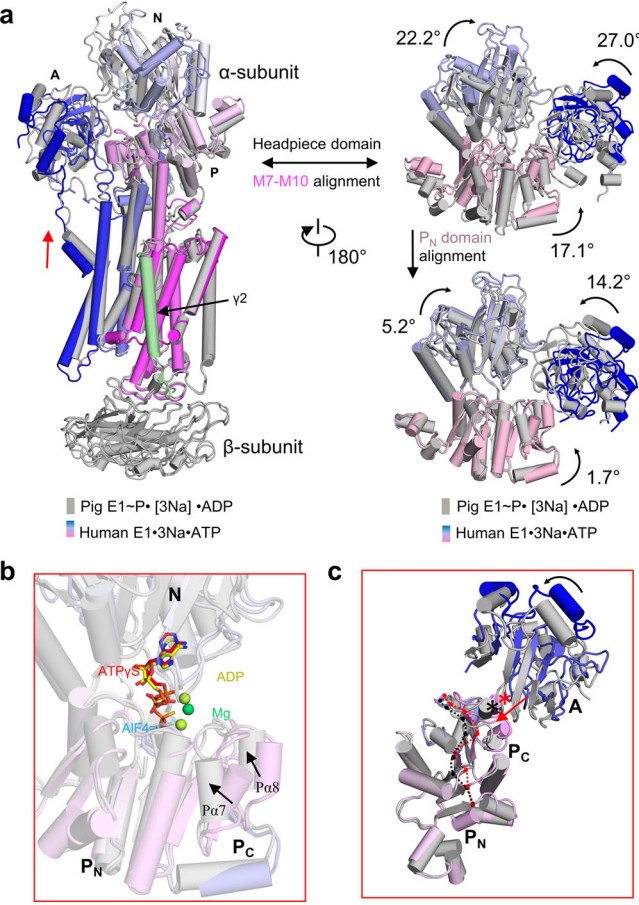

**Fig. 4 P domain conformation change causes cytoplasmic gate closure.**
**a** Structural comparison of the E1·3Na·ATP (blue to magenta) with
E1~P·[3Na]·ADP (grey) states with the movement of the M1 helix and
cytosolic headpiece shown. The structures are superimposed with M7–M10
or the P$_N$ domain (598–710 residues). Red arrows represent closing of the
M1 sliding door. **b, c** The structures are superimposed with the P$_N$ domain
(598–710 residues) between the E1·3Na·ATP and E1~P·[3Na]·ADP states.
The ATP (red) and ADP (yellow) binding pockets between the N and P
domains are shown in (**b**). The P$_N$ and P$_C$ half domains have a bent
conformation in the E1·3Na·ATP state (indicated by the red dashed line) or
an unbent confirmation in the E1~P·[3Na]·ADP state (indicated by the black
dashed line), respectively. The A domain sitting on a helix (Pα8) are
denoted with asterisk in E1·3Na·ATP (red) and in E1~P·[3Na]·ADP (black).
Red arrow indicates movement of Pα8 during transition from
the E1·3Na·ATP state to the E1~P·[3Na]·ADP state.

## Discussion

More than 30 crystal structures of NKA have been published,
focusing mainly on three basic states (E1~P·[3Na]·ADP, E2P·2K
and E2·Pi·[2K]) in the Post-Albers cycle. Here, we presented three
cryo-EM structures of NKA in different states using recombinant
proteins. To explore the state of our structures, we calculated the
paired RMSD values of all NKA structures (Supplementary
Fig. 14). The calculation results revealed that E1·3Na·ATP and
E1·3Na in this work belong to the E1 state and are quite distinct
from the E1~P·[3Na]·ADP state. Most of the pre-existing struc-
tures (85%) belong to the E2 state and can be classified into
the E2·P and E2·Pi states. E2·[2K] in this work is more similar to
the E2·Pi state than to the E2·P state. Therefore, our work has
revealed three conformations representing different states in the
Post-Albers cycle, for which no structural information has been
reported before.

NKA used for crystallization was purified from an endogenous
source, such as kidney or shark rectal glands. However, crystal-
lization of NKA turned out to be particularly difficult[30], although
recombinant proteins were already used to study the other P2-
type members (SERCA[31] and H⁺,K⁺-ATPase[32]). Here, we
develop a suitable strategy for structural determination using a
heterologous expression system and pave the way for further
investigation of the NKA complex that will likely be difficult for
crystallography. Beyond this, the use of ATPγS to trap NKA in
the E1 state suggest that ATPγS is an ATP analogue that can
mimic a situation preceding NKA phosphorylation.

Our data provides the structural insights into the cytoplasmic
ion-exchange pathway of NKA. We speculate that a gating
mechanism takes place in the cytoplasm in two stages (Supple-
mentary Fig. 15 and Supplementary Movie 1). The first stage is
gate open towards the intracellular side during the transition
from the E2·[2K] state to the E1·ATP state. Then, two K⁺ are
released into the cytosol. At this time, the side chain of gating
residue Glu334 is likely disordered, indicating that the cyto-
plasmic gate appeared to be open as a consequence of Glu334
disorder. Thus, three Na⁺ are able to bind from the intracellular
side. It seems likely that sites I and II and the side chain of gating
residue Glu334 near the door are unstable when Na⁺ binds to
NKA according to EM density (Fig. 2 and Supplementary Fig. 7a)
and Na⁺ valence (Supplementary table 2). The second gating
stage at the cytoplasmic side is the gate closure during the tran-
sition from the E1·3Na·ATP state to the E1~P·[3Na]·ADP state
(Fig. 4 and Supplementary Fig. 8). The M1 sliding door is elevated
to close the cytoplasmic gate, accompanying the dramatically
conformational changes with closed headpiece and an unbent P
domain that was proposed in a recent publication about
SERCA[25]. The bending of the P domain is required for making
phosphoryl transfer possible. Intriguingly, our data imply that
ATPγS binding mode in hNKA is different from ACP in
the SERCA E1·2Ca·ATP state[18,33], but is similar to ACP in
the SERCA E2·ACP state[25] (Fig. 3 and Supplementary Fig. 9a).
ACP in E1·2Ca·ATP is able to induce P$_C$ domain in a similar
unbent conformation as AlF4⁻ and ADP do in NKA[13]. We found
that Asp376 side chain has an upright orientation, which is likely
induced by the interaction between γ phosphate of ATP and
catalytic Asp376, is very important for the gate closure
mechanism by causing the P$_C$ domain bending releasing motion
(Fig. 4b, c) by comparing ATPγS, ACP, and ADP binding
pockets. This may be the reason for the M1 sliding door closed
coupling NKA phosphorylation to Na⁺ binding site occlusion.

## Methods

**Protein expression and purification.** The full-length cDNA of human α1 (Gene
name: ATP1A1, GenBank accession number: BAA00061.1) was subcloned into pCAG
with N-terminal FLAG tag, β1 (Gene name: ATP1B1, GenBank accession number:
CAA27385.1) into pCAG with C-terminal Strep tag and no tag γ2 (Gene name:
FXYD2, GenBank accession number: AAG37906.1) into pCAG vector. For protein
expression, α1, β1 and γ2 were co-expressed in HEK 293F cells (Invitrogen) which
were cultured in SMM 293T-II medium (Sino Biological Inc.) at 37 °C under 5% CO₂
in a Multitron-Pro shaker (Infors, 130 rpm). To transfect one liter of cells, 3 mg of
polyethylenimines (PEIs) (Polysciences), 0.5 mg of the α1 plasmid, 0.5 mg of the β1
plasmid and 0.5 mg of the γ2 plasmid were preincubated with 50 mL fresh medium
for 15 min and added into cell culture whose cell density reached 2.0 × 10⁶/mL. After
48 h of transfection, cells were harvested by centrifugation at 3800 × g for 10 min and
resuspended in a buffer containing 25 mM Tris (pH 8.0), 3 mM MgCl₂ and 150 mM
NaCl or 100 mM KCl for E1 or E2 state respectively, mixture of three protease
inhibitors, aprotinin (1.3 μg/ml, AMRESCO), pepstatin (0.7 μg/ml, AMRESCO) and
leupeptin (5 μg/ml, AMRESCO).

For protein purification, after incubating with 1.5% (w/v) n-dodecyl β-d-
maltoside (DDM, Anatrace) supplemented with 0.1% (w/v) cholesteryl
hemisuccinate tris salt (Anatrace) at 4 °C for 2 h, cells were centrifuged at
18,000 × g for 1 h to remove cell debris. The supernatant was loaded onto anti-
FLAG M2 affinity resin (Sigma). The resin was washed with the wash buffer
containing 25 mM Tris (pH 8.0), 3 mM MgCl2,150 mM NaCl or 100 mM KCl,
0.04% GDN (w/v), following by protein eluted with wash buffer plus 0.2 mg/mL

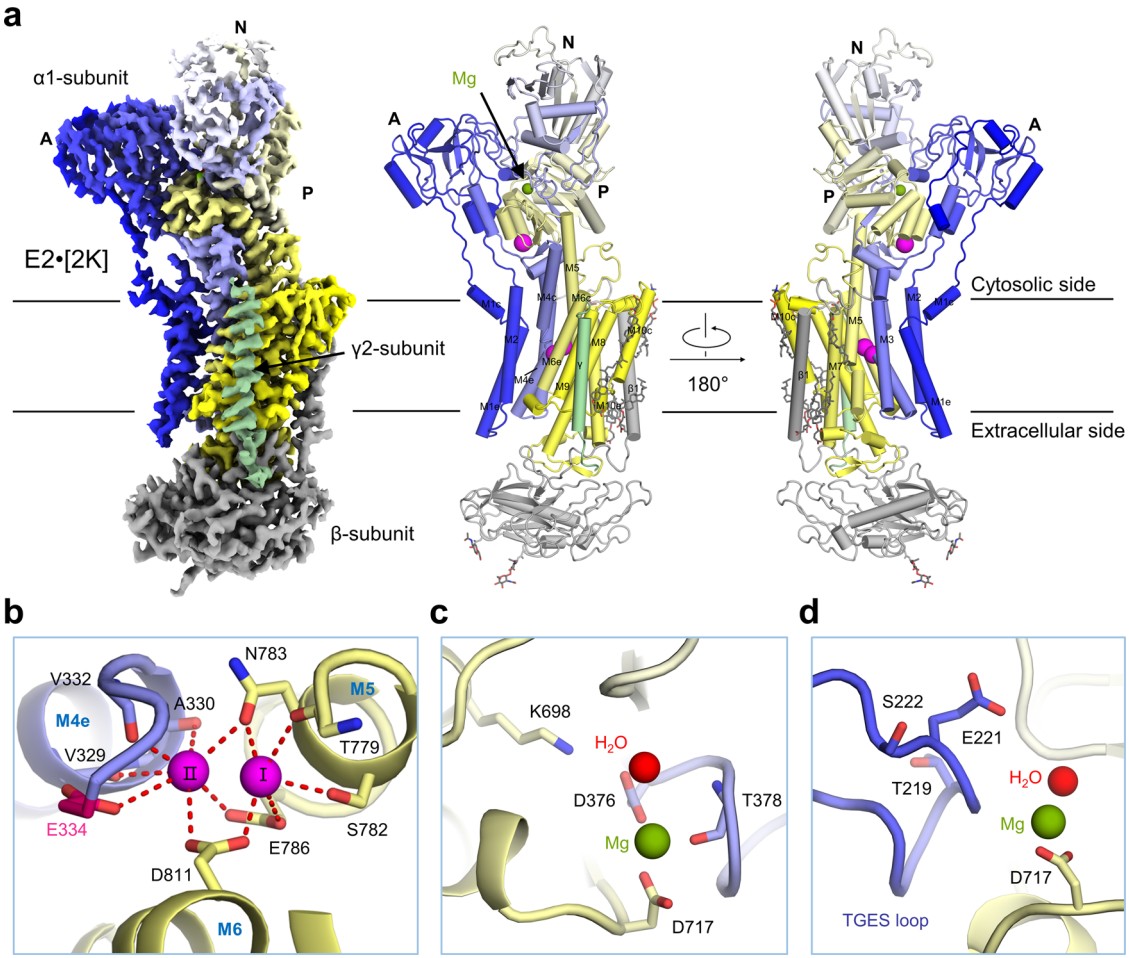

**Fig. 5 Overall structure of E2·2[K]. a** The overall cryo-EM map of E2·2[K] is shown in the left panel, and two perpendicular views of the overall structure are shown in the middle and right panels. The α1 subunit is rainbow coloured from blue for N-terminus to yellow for the C-terminus; the β1 subunit is coloured grey; the γ2 subunit is coloured pale green. **b** In the E2·2[K] state, two $K^+$ are bound to transmembrane $K^+$ binding sites (I and II). **c** $Mg^{2+}$ stabilizes phosphorylation site (Asp376). **d** The hallmark $^{219}$TGES motif in the A domain is located near $Mg^{2+}$. $K^+$ are coloured magenta; $Mg^{2+}$ is coloured olive green; and water molecules are coloured red. The glycosylation moieties are shown as sticks. e extracellular, c cytosolic, M transmembrane helix.

FLAG peptide. Then the protein complex was subjected to size-exclusion chromatography (Superose 6 Increase 10/300 GL, GE Healthcare) in buffer containing 25 mM Tris (pH 8.0), 3 mM MgCl2,150 mM NaCl or 100 mM KCl and 0.02% GDN. The peak fractions (14.5–15 ml) were collected and concentrated for EM analysis.

**Cryo-EM sample preparation and data acquisition**. The purified hNKA (α1β1γ2 complex) was concentrated to ~ 9 mg/mL and incubated with 1 mM ATP analogue (adenosine 5′-O-(3-thio) triphosphate, ATPγS) in Na+ buffer 1 h for E1·3Na·ATP state, absence of ATPγS for E1·3Na and in K+ buffer for E2·[2 K] respectively, before being applied to the grids. Aliquots (3.3 μL) of the protein complex were placed on glow-discharged holey carbon grids (Quantifoil Au R1.2/1.3). The grids were blotted for 3 s or 3.5 s and flash-frozen in liquid ethane cooled by liquid nitrogen with Vitrobot (Mark IV, Thermo Fisher Scientific). The cryo grids were transferred to a Titan Krios operating at 300 kV equipped with Gatan K3 Summit detector and GIF Quantum energy filter. Movie stacks were automatically collected using AutoEMation 2[34], with a slit width of 20 eV on the energy filter and a defocus range from −1.2 to −2.2 μm in super-resolution mode at a nominal magnification of ×81,000. Each stack was exposed for 2.56 s with an exposure time of 0.08 s per frame, resulting in a total of 32 frames per stack. The total dose rate was approximately 50 e-/Å2 for each stack. The stacks were motion corrected with MotionCor 2.1.1.0[35] and binned 2-fold, resulting in a pixel size of 1.087 Å/pixel. Meanwhile, dose weighting was performed[36]. The defocus values were estimated with Gctf 1.06[37].

**Data processing**. Particles were automatically picked from manually selected micrographs using Relion 3.0.6[38–41]. After 2D classification, good particles were selected and generated an initial 3D reference, then subjected to the global angular searching 3D classification using the initial model with only one class. For each of the last several iterations of the global angular searching 3D classification, a local angular searching 3D classification was performed, during which the particles were

classified into 4 classes. Non-redundant good particles were selected from the local angular searching 3D classification. Then, these selected particles were subjected to multi-reference 3D classification, local defocus correction, 3D auto-refinement and post-processing.

The 2D classification, 3D classification and 3D auto-refinement were performed with Relion 3.0.6. The resolution was estimated with the gold-standard Fourier shell correlation 0.143 criterion[42] with high-resolution noise substitution[43].

**Model building and structure refinement**. The atomic models of the hNKA were built based on the corresponding cryo-EM maps. A Chainsaw[44] model of the hNKA was first obtained using the previous structure of the sodium–potassium pump (PDB ID: 3KDP) as a template, which were further manually built in Coot 0.8.2[45]. Each residue was manually checked with the chemical properties considered during model building. Structure real space refinement was performed with Phenix 1.14[46] with secondary structure and geometry restraints to prevent structure overfitting. To monitor the overfitting of the model, the model was refined against one of the two independent half maps from the gold-standard 3D refinement approach. Then, the refined model was tested against the other map. Statistics associated with data collection, 3D reconstruction and model refinement can be found in Table S1. Structure analysis and figures were performed or prepared with PyMOL 2.1.0 (www.schrodinger.com/pymol), Chimera 1.15[47] or ChimeraX 1.1[48].

**Reporting summary**. Further information on research design is available in the Nature Research Reporting Summary linked to this article.

## Data availability
The data that support this study are available from the corresponding author upon reasonable request. The atomic coordinates and cryo-EM density maps of E1·3Na·ATP,

E1·3Na and E2·[2K] (PDB codes: 7E21 [https://doi.org/10.2210/pdb7E21/pdb], 7E1Z [https://doi.org/10.2210/pdb7E1Z/pdb] and 7E20 [https://doi.org/10.2210/pdb7E20/pdb]; whole maps: EMD-30949, EMD-30947 and EMD-30948 have been deposited in the Protein Data Bank and the Electron Microscopy Data Bank, respectively.

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

## Acknowledgements

We thank the cryo-EM facility, the high-performance computing centre and the mass spectrometry & metabolomics core facility of Westlake University for providing the technical supports. This work was supported by the National Key R&D Program of China (2020YFA0509300) and the National Natural Science Foundation of China (projects 31800139, 31971123).

## Author contributions

Q.Z. conceived the project. Q.Z. and Y.G. designed the experiments. Y.G., Y.Z. and R.Y. performed the experiments. F.Y., B.H., L.W. and X.C. contributed to the figures and movie. Y.S. contributed to the Na+ valence calculation. All of authors contributed to the data analysis. Q.Z. and Y.G. wrote the manuscript.

## Competing interests

The authors declare no competing interests.
