## [Peer Review File · Nature Communications]

Cryo-EM structures of recombinant human sodium-potassium pump determined in three different statesREVIEWER COMMENTS

Reviewer #1 (Remarks to the Author):

Guo et al reported cryo-EM structures of a human sodium pump reaching to 3.1Å resolution, sufficiently good enough for the model building. Among them, two out of three structures adopt in the cytoplasmic-open E1 state in which three sodium ions are bound, but not occluded, providing the first structural evidence for the cytoplasmic full-open gating achieved mostly by the unusual conformation of Glu residue at TM4 unwinding. Observed different positioning of Na⁺ in E1Na and E1NaATP states are also interesting, and may capture a transition state of Na⁺ acquisition from the cytoplasmic solution.

This reviewer agrees that the findings by Guo et al is a valuable contribution to the field, as such these results must be precisely evaluated. The reviewer hopes that the manuscript could be improved by a thoughtful revision, especially for the several issues as described below, and in a separate PDF file provided.

Major comments

Order of presentation. Naturally, Na-bound E1 state occurs before E1NaATP state. For example, in Fig 4bc, however, NaE1ATP comes in b and E1Na follows in c (cropped area in these figures must be the same to allow the audience precise comparison of Na⁺-binding position). Not only in figures, this order sometimes changes in the text as well, which is confusing.

In the PDF file, I, II and III are unreadable. These may be unintentional errors during pdf conversion, but critical for discussion which Na⁺ is indicated in the manuscript. Authors should be more careful for proofread of their produced files.

This reviewer found some critical problems regarding cytoplasmic-open gate structure, especially E334 side chain conformation. Details can be found in the PDF provided (E1state_comment.pdf).

The reason why cytoplasmic gate is fully open in E1-like state is because of an unusual conformation of E334 in TM4. This observation is not well focused in the manuscript. Author should discuss, the molecular mechanism of why E334 in these E1-type structures can take such a cytoplasmic full-open conformation. In NKA and even SERCA, so far not a single structure having such a conformation of E334 (E309 in SERCA) has been reported in my knowledge. In the E1Ca state of SERCA (Toyoshima et al., 2000, pdb 1su4), despite no backing support by TM1, E309 is facing to Ca²⁺ and contribute to the cation coordination. This is rather more significant in the E1Mg state (3w5b), in which TM1 is far away from E309, and apparently no support from TM1-2 residues, but E309 is still facing to the Ca²⁺ at site II.

In NKA E1-AIF-ADP state, Leu97 and Phe93 in TM1 support E327 to occlude Na⁺ in the cation site. Such atomic detail is not described or even not presented in the figures. Showing lateral helix movement in Suppl fig 5 is insufficient, at least side chains around E334 and cation-binding site is required to explain molecular mechanism of “open” conformation of E334 side chain.

Related, if Fig.4f, main chain trace (ribbon) in E334 significantly differs (2~3Å?) in E1NaATP and E1AIF-ADP. But the difference is hardly seen, rather these look almost the same, in Suppl fig 5c. The discrepancy must be confirmed.

Related to above comment, EM density map around E334 and Na⁺ ions must be presented in detail. EM densities in a Suopl Fig are too small and the density responsible for E334 is hardly seen. Moreover, its molecular details (e.g., comparison of the TM4 conformation between E1 states in this work and E1P-ADP state crystal structure) should be presented. Figs.4ef are obviously insufficient to explain molecular details of TM4 unwinding and E334 conformation.

To make another highlight (Na⁺ binding) more reliable, EM densities must be presented not only around Na⁺, but also coordinating residues. It is understandable that only showing densities around 3~4Å from Na⁺ is easy to recognize, but seems a bit unfair. Some of the densities are not spherical in the figure, and therefore this reviewer is skeptical whether the EM density resolve bound three Na⁺ at the cation-binding site (and this was true when I look at map files provided by the author). By showing Na⁺ density with coordinating residues, and surrounding noises as well, audience can judge how much reliable the presented structural data is to describe the Na⁺ binding position.

Related, author should estimate valences at the cation-binding site as presented in Shinoda et al. 2009 and Kanai et al. 2013. The densities that authors assigned as Na⁺ could be Na⁺, but also could be other ion or perhaps water. Providing valence value for each cation makes Na⁺-binding much reliable and convincing.

Fig.3b showed a coordination of ADP and AIF in the previously published structure. But binding details of ATPgammaS is missing, and only their comparison is shown in the following fig3c, in which surrounding residues are not presented at all. It was reported for SERCA E2-ACP state, D369 conformation is a key for the P domain bending (Kabashima et al, 2020, PNAS). In this respect, it is important to show which side D369 is facing.

In Fig. 5cd, two Mg²⁺ ions in close proximity, without having anionic groups such as phosphate or its analog (e.g. AlF₄⁻ or MgF₄⁻) in E2[2K] state is unlikely. Because one of the Mg²⁺ modeled is close to Asp376, one is probably SO₄⁻ or Pi carry-over during purification? Alternatively, EM density may be a summation of two different binding site. Valence estimation for Mg²⁺ may be helpful to confirm whether Mg²⁺ is capable for binding.

Minor comments

In Fig 4ef M1e is M2.

Fig4bcd, and Fig4ef, adjust magnification of the displayed items, so that these figures become easy to compare.

Site III Na⁺ looks buried in the protein. Is it possible to show the dimension of Na⁺-binding cavity much clearer using e.g. programs HOLE or CAVE?

In some figures, side chain is split from the main chain, probably because of cartoon representation of the model.

Including site I,II,III, careful proofread is needed.

Kazu Abe

There are some serious concerns about E1-related structures.

1. E334 conformation

The main reason why cytoplasmic gate opens in E1Na and E1Na-ATP states is an unusual conformation of E334 at TM4 unwinding. Among P2-ATPase structures (including NaK, SERCA and HK), not a single structure has been reported such an “open” conformation of E334. Although it is obviously expected from E1Ca (1su4) state that the route of Na⁺ dissociation into the cytoplasmic solution is via V-shaped structure of cytoplasmic portion of TM1-2, E309 (SERCA) is still facing to the cation-binding site and prevents full-opening of cation binding site to the cytoplasmic solution. Therefore, present structures are important piece, if these are really so.

As authors kindly provide unpublished maps, I could evaluate them precisely. And I found some serious problems regarding E334 conformation and the presentation of Na⁺-binding site in the manuscript.

Figure 1 EM density map and models around TM4 of E1Na state.

As shown in Fig.1, the side chain conformation of E334 cannot be determined from this EM density map. In the manuscript, there are no description about this poor EM density around E334. Authors showed EM densities of TM helices in suppl fig 4. By looking at TM4 density author provided, E334 side chain is kept conveniently hidden by looking at from this particular direction. I feel some intention to hide this poor density of E334 and rather some malicious in this figure. Because obviously the most important residue in this study is E334, but author just mark P333 at this position, and no indication about E334.

However, it is also true that the EM density does not indicate that E334 conformation is like usual “closed” one. The observed poor EM density suggests that only E334 side chain is disorder and does not averaged as EM density in this data set, because other residues are well resolved. Authors must describe about it and present clearly how E334 looks like in the EM map.

Situation is rather serious in E1NaATP state structure.

Fig. 2A EM density and model of E1Na-ATP state at high contour level

As is in E1Na state, EM density of E334 in E1Na-ATP state is also poor, as seen in Fig.2.

Fig. 2B EM density and models of E1Na-ATP at lower contour level

In the lower contour level, “open” conformation E334 side chain apparently covers by the EM density as in Fig2B. But as seen in the map, it is very noisy and the density could also be a noise. In this contour level, E334 could also be modeled as usual “closed” conformation, as seen in Fig. 2C

Fig. 2C EM density of E1Na-ATP at lower contour level, with an alternative model

Accordingly, in the present map, E334 conformation does not explicitly determined in both E1Na state and E1Na-ATP state. I think it is unfair neither to present EM density clearly enough to show E334, nor describe it in the manuscript. Above information must be included in the manuscript so that audience could judge it.

2. Na⁺-binding sites

Similar to E334 EM density, presentation of EM densities for bound Na⁺ ions are very unfair.

Fig3A Na⁺ ions in E1Na state

Fig3B Na⁺ ions in E1Na-ATP state

Fig3C Na⁺ ions in E1Na-ATP state at low contour level

The separate EM densities for Na⁺ ions that authors provided in Figure 4bc are obviously only showing 3~4Å cut-off from Na⁺ position, as seen in their non-spherical densities. Especially site III Na⁺ does not be a separate density, connecting to the surrounding residues. The real EM map is something like Fig3C in this PDF. There are many noises in the EM map. If authors state these densities are simply noise, author could not model E334 as an open conformation. Therefore, the way of presentation is somewhat convenient or biased throughout the manuscript. Space limitation is not the reason because there are plenty of supplementary items to show raw data including EM maps.

3. ATP_γS structure

Another interesting finding of this work is P-domain bending, although it has already reported in SERCA (e.g. Kabashima et al., 2020, PNAS). This should be compared and discussed in the manuscript.

In SERCA, E1-AMPPCP and E1AIF-ADP structures are indistinguishable. This is because gamma phosphate of AMPPCP reaches to the catalytic Asp, just like an AIF in the E1AIF-ADP state. In these structures of SERCA, and therefore E1AIF-ADP state of NKA as well, side chain conformation of Asp is “upright”, to interact with terminal phosphate or AIF, thus distinct to that observed in E2-ACP state of SERCA (Kabashima et al). In the ATP_γS structure of this work, D376 conformation clearly belongs to the latter case (SERCA E2-ACP type); D376 is facing to K698 (Fig4 in this PDF), and this interaction seems to provide a bending

state of the P domain different from that in E1AIF-ADP state of NKA. Detailed coordination of AIF-ADP in Figure 3b is not necessarily presented in the manuscript. Rather, ATPgS coordination must be shown in the manuscript (because authors determined it in this work), and should be compared to other reported structures including above mentioned NKA and SERCA, to show P domain bending mechanism as described above. In Figure 3c, P-domain bending motion is recognizable, but distinct Asp376 conformation cannot be seen. This reviewer think that this is the first example of ATPgammaS-bound P2-type ATPase structure. The observed experimental and structural fact that the ATPgS mimics a reaction state preceding E1P, because its terminal thio-phosphate does not interact to Asp, provides another option, other than AMPPCP, for the structural analysis of P-type ATPases, and therefore important contribution to the field.

Fig. 4 ATPgS and Asp376. Asp376 is facing to the Lys698 within a salt-bridge distance. At least, does not facing to the gamma-thio-phosphate side of ATPgS.

Reviewer #2 (Remarks to the Author):

Recombinant human Na,K-ATPase was overexpressed in HEK cells, isolated and purified to study its structure by cryo-EM technique. Choosing appropriate substrate conditions in the buffer, the proteins were trapped in three different states of the pump cycle: (1) E1·3Na, stabilized by 150 mM Na⁺ and 3 mM Mg²⁺; (2) E1·3Na·ATP, stabilized by 150 mM Na⁺, 3 mM Mg²⁺ and 1 mM ATPγS; (3) E2·[2K], stabilized by 100 mM K⁺ and 3 mM Mg²⁺. So far no high-resolution structures were published in these unphosphorylated E1 and E2 conformations of the pump. Both new E1 structures represent states in which the ion-binding sites are accessible from the cytoplasmic side and the chosen high Na⁺ concentration made sure that virtually all three ion sites were occupied permanently during structure determination. The main difference between both states E1·3Na and E1·3Na·ATP was that in the latter the nucleotide-binding site was occupied by the 'non-hydrolysable' ATP analogue, adenosine 5'-O-(3-thio) triphosphate. The third structure investigated was an unphosphorylated E2 state with 2 K⁺ occluded. This is the preferential state when the enzyme is incubated in the E1 conformation with saturating concentrations of K⁺.

These structures are of considerable interest since they may contribute to an enhanced understanding of the molecular mechanism of cytoplasmic ion binding and release. So far reliable mechanistic proposals for these reaction steps were derived only from studies of the ion-binding kinetics in biophysical and biochemical experiments.

According to the detailed description in the Supplementary Materials the determination of all three structures was performed with state-of-the-art equipment and analysis techniques. Using a previously published template, a refined and reliable 3D reconstruction was obtained for all three presented structures with resolutions of 3.1 Å to 3.4 Å.

By comparing structures from both E1 states, E1·3Na and E1·3Na·ATP, unsurprisingly it was found that they were "virtually identical" (line 80) with minor differences in the nucleotide binding pocket, which was empty in one case and occupied by the ATP analogue in the other. These minor changes, nevertheless, seem to affect slightly, via transmembrane helices M4 and M5, ion-binding sites I and II, and therefore, the position of the Na⁺ ions coordinated therein.

The determined structure of the K⁺-occluded E2 state showed no significant difference to the previously published structure of the K⁺-occluded state preceding in the pump cycle that has a phosphate attached to the P domain. The agreement holds for both the position of the K⁺-binding sites and the phosphorylation site. When compared to the E1·3Na·ATP state, obtained by other authors from pig enzyme, the differences in the arrangement of the cytoplasmic N, P

and A domains are obvious, but obviously besides the position of helices M1 and M2 only minor shifts were found at the location of the ion sites (Suppl. Fig. 8b).

The two E1 structures were used to develop a mechanistic proposal by comparison of the new structures with the previously published structure of the $E1\sim P\cdot[3Na]\cdot ADP$ state, which follows in the Post-Albers pump cycle after $E1\cdot 3Na\cdot ATP$ as next state (and served also as template). While the new E1 structures represent states in which the access pathway from the cytoplasm to the ion-binding sites is open, the 3 Na⁺ ions are occluded in the state $E1\sim P\cdot[3Na]\cdot ADP$. By comparing both structures evidence for the occlusion mechanism may be derived. A prominent difference between the $E1\cdot 3Na\cdot ATP$ and $E1\sim P\cdot[3Na]\cdot ADP$ structures is the movement of the M1 (M1c & M1e) helix towards the cytoplasmic side when enzyme phosphorylation occurred according to the Post-Albers cycle. This movement, here named “sliding door”, must be caused by rearrangement of the N/P/A domains of the cytoplasmic “headpiece” of the Na,K-ATPase as consequence of the phosphate transfer from ATP to the P domain. This suggested concept is descriptive and intriguing. To be convincing, however, additional proof is required. One line of argument can be based on results already provided by the authors: The structure of the occluded $E2\cdot[2K]$ state should exhibit also a closed “sliding door”. An additional figure presenting a comparison of the arrangement of the TM helices in the membrane domain of both states, $E1\sim P\cdot[3Na]\cdot ADP$ and $E2\cdot[2K]$, which form the ion sites, the access structure, and the sliding door, would reveal whether the position of the “gate” concurs. If so, it supports the concept, if not, additional arguments have to be presented to justify the sliding-door concept.

Supportive would be also a presentation and comparison of similar arrangements of the M1 helix of SERCA in the open and occluded state. In the presented supplementary Figure 9 from superficial inspection at least the obvious differences of the position of the TMs seem to be not significant in the open state $E1\cdot 2Ca\cdot ATP$ and $E1P\cdot ADP\cdot 2Ca$, which has to be an occluded state. A detailed comparison of both states can easily be derived from available structures. Due to the overall similarities in structure and function of Na,K-ATPase and Ca-ATPase, the sliding-door mechanism should be expected to be a common feature.

Another claim, concerning the Na⁺-binding sequence (lines 132-135), should be reconsidered. There are only structures available in which all three sites are occupied. Why is it “obvious” (line 132) that site III is filled in first? The fact that “channel” is too narrow between the cytoplasmic aqueous phase and the innermost site III so that a Na⁺ cannot pass ions already localized in site I or II, is not sufficient. In similarly narrow cation channels a so-called single-file mechanism was proven to describe ion translocation: A concerted movement of the ions along the sequence of sites in the channel takes place by which the next ion entering the narrow channel structure “pushes” or “persuades” the ion currently present in the immediate site to move on to the next site and allows the next ion to be bound. With respect to the Na,K-ATPase, therefore, the occupation sequence for the three Na⁺ sites could well be: $(II, _, _) \rightarrow (II, I, _) \rightarrow (II, I, III)$. In this case site III would be filled last. Can you exclude this scenario or what makes it less “obvious”? This pattern was favored in the

discussion of kinetic studies that traced charge movements during ion binding. The fact that your proposal is in agreement with the (also unproven) hypothesis in Ref. 14 is hardly convincing.

The language of the manuscript needs urgently extensive improvements by a native English-speaking person!

Minor points

lines 54-55: The meaning of this sentence is obscure. Since you do not have a valid structure before Na⁺ binding, how do you know that there are “large conformational changes during Na⁺ access to the ion-binding sites”?

lines 99-100: “During the E1·3Na/E1·3Na·ATP to E1~P·[3Na]·ADP transition, Na⁺-binding sites move toward the depth of cation binding cavity (Fig. 4e, f).” However, according to these figures a noteworthy ion movement occurred only between E1·3Na and E1·3Na·ATP but not between E1·3Na·ATP and E1~P·[3Na]·ADP. This visible fact is confirmed by the supplementary video. And the claim of a (notable) movement “toward the depth of cation binding cavity” may be true only for ion II in E1·3Na (Fig. 4e).

line 112: What is the meaning of “ADP has a smaller pocket by the tilt with N and P domain ...” Does ADP ‘require a smaller pocket’ or ‘induces a reduction of the pocket’? Is the tilt of the N and P domain cause (as this sentence may claim) or effect?

lines 210-214: The E1·3Na·ATP and E1·3Na structures DO NOT reveal a cytoplasmic gating mechanism correlated with the ATP-dependent Na⁺-binding site remodeling. If anything, their comparison with the E1~P·[3Na]·ADP structure does.

Supplementary Materials

line 29: How many/which peak fractions were collected?

line 94, Supplementary Fig. 2: Correction necessary, panel “f” should be “d”, and panel “i” should be “e”.

Reviewer #3 (Remarks to the Author):

The paper by Guo et al., entitled “Cryo-EM structures of the human sodium-potassium pump revealing the gating mechanism on the cytoplasmic side.”

Describe the cryo-EM structure of the human Na, K ATPase expressed recombinantly in HEK cells and determined structural in the E1 state with three sodium ions bound in both the presence of an ATP analogue and without. The authors also present the cryoEM structure of the E2 state bound with two potassium ions. The study represents a milestone on several levels. It is the first cryo-EM structure of recombinantly produced Na, K ATPase. It will likely pave the way for understanding the several disease mutants known for this important enzyme and perhaps describing how many of the known drugs affect the Na, K ATPase, which likely will not be possible for crystallography, i. e the palytoxin bound structure.

The paper is exceptionally well written and in easy term describe how the sodium and potassium ions are bound, and confirm the position with the known crystal structures and thus will likely become the method of choice for future structural endeavours for members of the P-type ATPase family.

What seems to be missing is a description of the lipids, in particular the cholesterol site present in the Na, K -ATPase. At this resolution, it should be visible, and what of other lipids bound in the transmembrane region?

As this paper describes a complete description of Na, K ATPase from a recombinantly expressed protein, a description of the lipid environment should be included.

The title is overselling the paper. Only structural data is presented for a putative gating mechanism and would require functional data to be included. The title should be reduced to Cryo-EM structures of recombinant human sodium-potassium pumps determined in two different states.

We thank the reviewers for their thoughtful and constructive comments and respond to each point here:

Reviewer #1(Remarks to the Author):

Guo et al reported cryo-EM structures of a human sodium pump reaching to 3.1 Å resolution, sufficiently good enough for the model building. Among them, two out of three structures adopt in the cytoplasmic-open E1 state in which three sodium ions are bound, but not occluded, providing the first structural evidence for the cytoplasmic full-open gating achieved mostly by the unusual conformation of Glu residue at TM4 unwinding. Observed different positioning of Na⁺ in E1Na and E1NaATP states are also interesting, and may capture a transition state of Na⁺ acquisition from the cytoplasmic solution.

This reviewer agrees that the findings by Guo et al is a valuable contribution to the field, as such these results must be precisely evaluated. The reviewer hopes that the manuscript could be improved by a thoughtful revision, especially for the several issues as described below, and in a separate PDF file provided.

We thank this reviewer's support. We made a complete overhaul of the manuscript, especially for the critical issues commented or suggested by this reviewer.

Major comments

1. Order of presentation. Naturally, Na-bound E1 state occurs before E1NaATP state. For example, in Fig 4bc, however, NaE1ATP comes in b and E1Na follows in c (cropped area in these figures must be the same to allow the audience precise comparison of Na⁺-binding position). Not only in figures, this order sometimes changes in the text as well, which is confusing.

We thank this reviewer for pointing this out. Presentation order of E1·3Na and E1·3Na·ATP states in our manuscript have been modified and improved carefully. We have changed the order of E1·3Na and E1·3Na·ATP states in the original Fig 4bc (replaced by Fig2 now). We modified the improperly size and cropped area in all images of the revised manuscript.

2. In the PDF file, I, II and III are unreadable. These may be unintentional errors during pdf conversion, but critical for discussion which Na⁺ is indicated in the manuscript. Authors should be more careful for proofread of their produced files.

We thank this reviewer. We have checked our PDF version carefully.

3. This reviewer found some critical problems regarding cytoplasmic-open gate structure, especially E334 side chain conformation. Details can be found in the PDF provided (E1state_comment.pdf).

We thank this reviewer. We accepted these suggestions and answered these critical questions in ‘Reviewer #1 attachment (PDF file)’ part.

4. The reason why cytoplasmic gate is fully open in E1-like state is because of an unusual conformation of E334 in TM4. This observation is not well focused in the manuscript. Author should discuss, the molecular mechanism of why E334 in these E1-type structures can take such a cytoplasmic full-open conformation. In NKA and even SERCA, so far not a single structure having such a conformation of E334 (E309 in SERCA) has been reported in my knowledge. In the E1Ca state of SERCA (Toyoshima et al., 2000, pdb 1su4), despite no backing support by TM1, E309 is facing to Ca^{2+} and contribute to the cation coordination. This is rather more significant in the E1Mg state (3w5b), in which TM1 is far away from E309, and apparently no support from TM1-2 residues, but E309 is still facing to the Ca^{2+} at site II.

We thank this reviewer for this insightful comment. We improved the resolution of the E1·3Na and E1·3Na·ATP states to 3.2 Å and 2.9Å, respectively. We prefer the orientation of the E334 side chain does not face site II Na^+ to coordinate with it in the cryo-EM map of E1·3Na·ATP, considering the environment surrounding E334 (Figure R1). We regarded the reason why E334 side chain orientation in NKA is different from E309 of SERCA is Na^+ binding is not properly formed or stabilized at site II, although details were not investigated. We speculate side chain of E334 is unstable for the formation of Na^+ -coordination. We also discuss this in the “ Na^+ -binding sites and an open cytoplasmic accessible pathway” section in the revised manuscript.

Figure R1. Glu334 side chain orientation in the cryo-EM map of the E1·3Na·ATP state.

5. In NKA E1-A1F-ADP state, Leu97 and Phe93 in TM1 support E327 to occlude Na^+ in the

cation site. Such atomic detail is not described or even not presented in the figures. Showing lateral helix movement in Suppl fig 5 is insufficient, at least side chains around E334 and cation-binding site is required to explain molecular mechanism of “open” conformation of E334 side chain.

We thank this reviewer for this suggestion. When M1 sliding door is open, the two supporting residues Phe100 and Leu104 of hNKA (corresponding to residues Phe93 and Leu97 in pig NKA) shift away from Glu334 in the E1·3Na·ATP state and do not participate in the stabilization of Glu334. The residues near E334 are shown in Supplementary Fig7 and Supplementary Fig9a-b, and also discussed in the revised manuscript.

6. Related, if Fig.4f, main chain trace (ribbon) in E334 significantly differs (2~3Å?) in E1NaATP and E1AIF-ADP. But the difference is hardly seen, rather these look almost the same, in Suppl fig 5c. The discrepancy must be confirmed.

We thank this reviewer for pointing this out. In Fig. 2e (instead of original Fig.4f), main chain trace (ribbon) in E334 significantly differs ~3 Å (Figure R2). The reason these look almost the same in Supplementary Fig. 9d (instead of original Suppl fig 5c) is that the model shows different angles.

Figure R2. Main chain trace of E334 significantly differs in the E1-Na-ATP (pink and blue) and E1-AIF-ADP (grey) states. The sodium ion is colored in yellow or teal in the E1-Na-ATP and E1-AIF-ADP states, respectively.

7. Related to above comment, EM density map around E334 and Na⁺ ions must be presented in detail. EM densities in a Supl Fig are too small and the density responsible for E334 is hardly seen. Moreover, its molecular details (e.g., comparison of the TM4 conformation between E1 states in this work and E1P-ADP state crystal structure) should be presented. Figs.4ef are obviously insufficient to explain molecular details of TM4 unwinding and E334 conformation.

We thank this reviewer for this suggestion. In revised manuscript, we add Supplementary Fig.7 and Supplementary Fig.9a-b to show the EM density around E334 and the TM4 unwinding for Na⁺ ion coordinating.

8. To make another highlight (Na⁺ binding) more reliable, EM densities must be presented not only around Na⁺, but also coordinating residues. It is understandable that only showing densities around 3~4Å from Na⁺ is easy to recognize, but seems a bit unfair. Some of the densities are not spherical in the figure, and therefore this reviewer is skeptical whether the EM density resolve bound three Na⁺ at the cation-binding site (and this was true when I look at map files provided by the author). By showing Na⁺ density with coordinating residues, and surrounding noises as well, audience can judge how much reliable the presented structural data is to describe the Na⁺ binding position.

We thank this reviewer for pointing this out. We have improved the resolution of the E1·3Na and E1·3Na·ATP states to 3.2 Å and 2.9Å, respectively. We add Supplementary Fig.7 to show the EM density around E334 and Na⁺ ions. Coordinating residues and surrounding noises are shown in the revised manuscript to help audience judge how reliable the data is.

9. Related, author should estimate valences at the cation-binding site as presented in Shinoda et al. 2009 and Kanai et al. 2013. The densities that authors assigned as Na⁺ could be Na⁺, but also could be other ion or perhaps water. Providing valence value for each cation makes Na⁺-binding much reliable and convincing.

We appreciate your kind consideration of our manuscript. We estimated the valences of several nonprotein densities at the Na⁺-binding site using the method from the paper “Valence screening of water in protein crystals reveals potential Na⁺ binding sites” in the E1·3Na and E1·3Na·ATP states. We summarized the results in two new tables (Supplementary Table 2 and Supplementary Table 3) in the revised manuscript.

10. In Fig. 5cd, two Mg²⁺ ions in close proximity, without having anionic groups such as phosphate or its analog (e.g. AlF₄⁻ or MgF₄⁻) in E2[2K] state is unlikely. Because one of the Mg²⁺ modeled is close to Asp376, one is probably SO₄⁻ or Pi carry-over during purification? Alternatively, EM density may be a summation of two different binding site. Valence estimation for Mg²⁺ may be helpful to confirm whether Mg²⁺ is capable for binding.

We thank this reviewer for pointing this out. We also improved the resolution of the cryo-EM map of E2·[2K] state to 2.7 Å resolution, in which the Mg²⁺ density is clearly seen at high

contour level as shown in the Figure R3 below. We changed one of the two Mg^{2+} ions to a water molecule.

Figure R3. Mg^{2+} ion near Asp376 in ATP binding pocket

Minor comments

11. In Fig 4ef M1e is M2.

Point taken. The original Fig.4 is replaced to Fig. 2 in the revised manuscript. We have corrected it to “M2” in Fig. 2.

12. Fig4bcd, and Fig4ef, adjust magnification of the displayed items, so that these figures become easy to compare.

Point taken. We have adjusted magnification of the displayed items in Fig. 2 (the original Fig.4 in previous manuscript).

13. Site III Na^+ looks buried in the protein. Is it possible to show the dimension of Na^+ -binding cavity much clearer using e.g. programs HOLE or CAVE?

Point taken. We have shown the cavity by HOLE in Supplementary Fig. 8.

14. In some figures, side chain is split from the main chain, probably because of cartoon representation of the model.

Point taken. We have corrected these figures.

15. Including site I,II,III, careful proofread is needed.

Point taken. We have checked our manuscript and the PDF version carefully.

Reviewer #1 attachment (PDF file):

There are some serious concerns about E1-related structures.

17. E334 conformation

The main reason why cytoplasmic gate opens in E1Na and E1Na-ATP states is an unusual conformation of E334 at TM4 unwinding. Among P2-ATPase structures (including NaK, SERCA and HK), not a single structure has been reported such an “open” conformation of E334. Although it is obviously expected from E1Ca (1su4) state that the route of Na⁺ dissociation into the cytoplasmic solution is via V-shaped structure of cytoplasmic portion of TM1-2, E309 (SERCA) is still facing to the cation-binding site and prevents full-opening of cation binding site to the cytoplasmic solution. Therefore, present structures are important piece, if these are really so. As authors kindly provide unpublished maps, I could evaluate them precisely. And I found some serious problems regarding E334 conformation and the presentation of Na⁺-binding site in the manuscript. As shown in Fig.1, the side chain conformation of E334 cannot be determined from this EM density map. In the manuscript, there are no description about this poor EM density around E334. Authors showed EM densities of TM helices in suppl fig 4. By looking at TM4 density author provided, E334 side chain is kept conveniently hidden by looking at from this particular direction. I feel some intention to hide this poor density of E334 and rather some malicious in this figure. Because obviously the most important residue in this study is E334, but author just mark P333 at this position, and no indication about E334. However, it is also true that the EM density does not indicate that E334 conformation is like usual “closed” one. The observed poor EM density suggests that only E334 side chain is disorder and does not averaged as EM density in this data set, because other residues are well resolved. Authors must describe about it and present clearly how E334 looks like in the EM map.

Situation is rather serious in E1NaATP state structure. As is in E1Na state, EM density of E334 in E1Na-ATP state is also poor, as seen in Fig.2. In the lower contour level, “open” conformation E334 side chain apparently covers by the EM density as in Fig2B. But as seen in the map, it is very noisy and the density could also be a noise. In this contour level, E334 could also be modeled as usual “closed” conformation, as seen in Fig. 2C

Figure 1 EM density map and models around TM4 of E1Na state.

Fig. 2A EM density and model of E1Na-ATP state at high contour level

Fig. 2B EM density and models of E1Na-ATP at lower contour level

Fig. 2C EM density of E1Na-ATP at lower contour level, with an alternative model

We thank this reviewer for these insightful comments. We did not hide any information on purpose. It is a common phenomenon that the cryo-EM density is relative poor for acidic residues Asp and Glu. This phenomenon is described by Bartesaghi et al. (Bartesaghi, A., Matthies, D., Banerjee, S., Merk, A. & Subramaniam, S. Structure of beta-galactosidase at 3.2-Å resolution obtained by cryo-electron microscopy. *Proceedings of the National Academy of Sciences of the United States of America* 111, 11709-11714, doi:10.1073/pnas.1402809111 (2014).)

We now improved the resolution to 3.2 Å and 2.9 Å in E1·3Na and E1·3Na·ATP states respectively, which allowed more reliable visualization of densities near E334 and Na⁺ ions. We can see the orientation of E334 side chain is not facing to site II Na⁺ in the new cryo-EM map of the E1·3Na·ATP state (Figure R1). We observed a full-opening cavity of cation binding site in NKA.

18. Na⁺-binding sites

Similar to E334 EM density, presentation of EM densities for bound Na⁺ ions are very unfair. The separate EM densities for Na⁺ ions that authors provided in Figure 4bc are obviously only showing 3~4Å cut-off from Na⁺ position, as seen in their non-spherical densities. Especially site III Na⁺ does not be a separate density, connecting to the surrounding residues. The real EM map is something like Fig3C in this PDF. There are many noises in the EM map. If authors state these densities are simply noise, author could not model E334 as an open conformation. Therefore, the way of presentation is somewhat convenient or biased throughout the manuscript. Space limitation is not the reason because there are plenty of supplementary items to show raw data including EM maps.

Fig3A Na⁺ ions in E1Na state

Fig3B Na⁺ ions in E1Na-ATP state

Fig3C Na⁺ ions in E1Na-ATP state at low contour level

Thank you for pointing this out. We have improved the resolution of the E1·3Na and E1·3Na·ATP states to 3.2 Å and 2.9 Å, respectively. In the new structures with higher resolution, only a few noises were shown around Na⁺ binding sites even at lower contour level in the E1·3Na·ATP state. But, in E1·3Na state, there are still many noises at the mouth of the open

accessible tunnel. The side chain of E334 is more flexible in E1·3Na state. We have added a figure showing the density details of Na⁺ binding sites in the supplementary materials Figure7a.

19. ATP γ S structure

Another interesting finding of this work is P-domain bending, although it has already reported in SERCA (e.g. Kabashima et al., 2020, PNAS). This should be compared and discussed in the manuscript.

In SERCA, E1-AMPPCP and E1AIF-ADP structures are indistinguishable. This is because gamma phosphate of AMPPCP reaches to the catalytic Asp, just like an AIF in the E1AIF-ADP state. In these structures of SERCA, and therefore E1AIF-ADP state of NKA as well, side chain conformation of Asp is “upright”, to interact with terminal phosphate or AIF, thus distinct to that observed in E2-ACP state of SERCA (Kabashima et al). In the ATP γ S structure of this work, D376 conformation clearly belongs to the latter case (SERCA E2-ACP type); D376 is facing to K698 (Fig4 in this PDF), and this interaction seems to provide a bending state of the P domain different from that in E1AIF-ADP state of NKA. Detailed coordination of AIF-ADP in Figure 3b is not necessarily presented in the manuscript. Rather, ATP γ S coordination must be shown in the manuscript (because authors determined it in this work), and should be compared to other reported structures including above mentioned NKA and SERCA, to show P domain bending mechanism as described above. In Figure 3c, P-domain bending motion is recognizable, but distinct Asp376 conformation cannot be seen. This reviewer think that this is the first example of ATP γ S-bound P2-type ATPase structure. The observed experimental and structural fact that the ATP γ S mimics a reaction state preceding E1P, because its terminal thio-phosphate does not interact to Asp, provides another option, other than AMPPCP, for the structural analysis of P-type ATPases, and therefore important contribution to the field.

Fig. 4 ATP γ S and Asp376. Asp376 is facing to the Lys698 within a salt-bridge distance. At least, does not facing to the gamma-thio-phosphate side of ATP γ S.

We followed this reviewer's suggestion and added a new paragraph in the results section "*The binding mode of ATP γ S*" to describe the ATP γ S coordination and the distinct Asp376 conformation in E1 states of NKA and the E2·ACP (7btj) and E1·2Ca²⁺·ACP (1T5S) states. This information were shown in Figure 3 and supplementary material Figure 10a of manuscript. We also show P domain bending mechanism in the results section "*The cytoplasmic gate closure coupled with ATP hydrolysis*" and add a diagram (Figure 4c) to show the P domain bending clearly.

We thank this reviewer for carefully reading our manuscript and for all the critical and constructive comments.

Reviewer #2 (Remarks to the Author):

Recombinant human Na,K-ATPase was overexpressed in HEK cells, isolated and purified to study its structure by cryo-EM technique. Choosing appropriate substrate conditions in the buffer, the proteins were trapped in three different states of the pump cycle: (1) E1·3Na, stabilized by 150 mM Na⁺ and 3 mM Mg²⁺; (2) E1·3Na·ATP, stabilized by 150 mM Na⁺, 3 mM Mg²⁺ and 1 mM ATP γ S; (3) E2·[2K], stabilized by 100 mM K⁺ and 3 mM Mg²⁺. So far no high-resolution structures were published in these unphosphorylated E1 and E2 conformations of the pump. Both new E1 structures represent states in which the ion-binding sites are accessible from the cytoplasmic side and the chosen high Na⁺ concentration made sure that virtually all three ion sites were occupied permanently during structure determination. The main difference between both states E1·3Na and E1·3Na·ATP was that in the latter the nucleotide-binding site was occupied by the 'non-hydrolysable' ATP analogue, adenosine 5'-O-(3-thio) triphosphate. The third structure investigated was an unphosphorylated E2 state with 2 K⁺ occluded. This is the preferential state when the enzyme is incubated in the E1 conformation with saturating concentrations of K⁺.

These structures are of considerable interest since they may contribute to an enhanced understanding of the molecular mechanism of cytoplasmic ion binding and release. So far reliable mechanistic proposals for these reaction steps were derived only from studies of the ion-binding kinetics in biophysical and biochemical experiments.

According to the detailed description in the Supplementary Materials the determination of all three structures was performed with state-of-the-art equipment and analysis techniques. Using a previously published template, a refined and reliable 3D reconstruction was obtained for all three presented structures with resolutions of 3.1 Å to 3.4 Å.

By comparing structures from both E1 states, E1·3Na and E1·3Na·ATP, unsurprisingly it was found that they were "virtually identical" (line 80) with minor differences in the nucleotide binding pocket, which was empty in one case and occupied by the ATP analogue in the other.

These minor changes, nevertheless, seem to affect slightly, via transmembrane helices M4 and M5, ion-binding sites I and II, and therefore, the position of the Na⁺ ions coordinated therein.

The determined structure of the K⁺-occluded E2 state showed no significant difference to the previously published structure of the K⁺-occluded state preceding in the pump cycle that has a phosphate attached to the P domain. The agreement holds for both the position of the K⁺-binding sites and the phosphorylation site. When compared to the E1·3Na·ATP state, obtained by other authors from pig enzyme, the differences in the arrangement of the cytoplasmic N, P and A domains are obvious, but obviously besides the position of helices M1 and M2 only minor shifts were found at the location of the ion sites (Suppl. Fig. 8b).

The two E1 structures were used to develop a mechanistic proposal by comparison of the new structures with the previously published structure of the E1~P·[3Na]·ADP state, which follows in the Post-Albers pump cycle after E1·3Na·ATP as next state (and served also as template). While the new E1 structures represent states in which the access pathway from the cytoplasm to the ion-binding sites is open, the 3 Na⁺ ions are occluded in the state E1~P·[3Na]·ADP. By comparing both structures evidence for the occlusion mechanism may be derived. A prominent difference between the E1·3Na·ATP and E1~P·[3Na]·ADP structures is the movement of the M1 (M1c & M1e) helix towards the cytoplasmic side when enzyme phosphorylation occurred according to the Post-Albers cycle. This movement, here named “sliding door”, must be caused by rearrangement of the N/P/A domains of the cytoplasmic “headpiece” of the Na,K-ATPase as consequence of the phosphate transfer from ATP to the P domain. This suggested concept is descriptive and intriguing.

We thank this reviewer’s support.

1. To be convincing, however, additional proof is required. One line of argument can be based on results already provided by the authors: The structure of the occluded E2·[2K] state should exhibit also a closed “sliding door”. An additional figure presenting a comparison of the arrangement of the TM helices in the membrane domain of both states, E1~P·[3Na]·ADP and E2·[2K], which form the ion sites, the access structure, and the sliding door, would reveal whether the position of the “gate” concurs. If so, it supports the concept, if not, additional arguments have to be presented to justify the sliding-door concept.

We thank this reviewer for this suggestion. We have performed this kind of comparison as this reviewer suggested (Figure 4). The structures of the E1~P·[3Na]·ADP and E2·[2K] states are aligned using TM7-TM10, which reveals a downward movement of M1 and M2 helices in the E2·[2K] state. This result supports the sliding door concept.

Figure R4 Structural comparison between the E1~P·[3Na]·ADP state with the E2·[2K] state

2.Supportive would be also a presentation and comparison of similar arrangements of the M1 helix of SERCA in the open and occluded state. In the presented supplementary Figure 9 from superficial inspection at least the obvious differences of the position of the TMs seem to be not significant in the open state E1·2Ca·ATP and E1P·ADP·2Ca, which has to be an occluded state. A detailed comparison of both states can easily be derived from available structures. Due to the overall similarities in structure and function of Na,K-ATPase and Ca-ATPase, the sliding-door mechanism should be expected to be a common feature.

Thank you for pointing this out. We now state it more clearly and add a new paragraph in “*The binding mode of ATP γ S*” for describing the advantages of using ATP γ S than AMPPCP or AMPPCP in E1·2Ca·ATP and E1P·ADP·2Ca structures of SERCA. The gamma phosphate of AMPPCP mimic Pi to stabilize SERCA structure in phosphorylated E1 conformation. However, the ATP γ S can mimic the real situation in E1·3Na·ATP state.

3.Another claim, concerning the Na⁺-binding sequence (lines 132-135), should be reconsidered. There are only structures available in which all three sites are occupied. Why is it “obvious” (line 132) that site III is filled in first? The fact that “channel” is too narrow between the cytoplasmic aqueous phase and the innermost site III so that a Na⁺ cannot pass ions already localized in site I or II, is not sufficient. In similarly narrow cation channels a so-called single-file mechanism was proven to describe ion translocation: A concerted movement of the ions

along the sequence of sites in the channel takes place by which the next ion entering the narrow channel structure “pushes” or “persuades” the ion currently present in the immediate site to move on to the next site and allows the next ion to be bound. With respect to the Na,K-ATPase, therefore, the occupation sequence for the three Na⁺ sites could well be: (II,_,_) → (II,I,_) → (II,I,III). In this case site III would be filled last. Can you exclude this scenario or what makes it less “obvious” ? This pattern was favored in the discussion of kinetic studies that traced charge movements during ion binding. The fact that your proposal is in agreement with the (also unproven) hypothesis in Ref. 14 is hardly convincing.

We agree with this reviewer’s suggestion. After consideration, we removed this part about Na⁺-binding sequence.

4.The language of the manuscript needs urgently extensive improvements by a native English-speaking person!

The language has been corrected and polished by English language editing from Springer Nature Author Services (authorservices.springernature.com).

Minor points

5.lines 54-55: The meaning of this sentence is obscure. Since you do not have a valid structure before Na⁺ binding, how do you know that there are “large conformational changes during Na⁺ access to the ion-binding sites” ?

We thank this reviewer to point it out. What we mean here is that NKA needs to open its cytosolic gate to release K⁺ ions and facilitate subsequent binding of Na⁺ ions from cytosolic side. To be more accurate, we changed this sentence to “*NKA undergoes large conformational changes during K⁺ release and Na⁺ access to the ion-binding sites and is occluded by cytoplasmic gate closing.*”

6.lines 99-100: “During the E1 • 3Na/E1 • 3Na • ATP to E1~P • [3Na] • ADP transition, Na⁺-binding sites move toward the depth of cation binding cavity (Fig. 4e, f).” However, according to these figures a noteworthy ion movement occurred only between E1 • 3Na and E1 • 3Na • ATP but not between E1 • 3Na • ATP and E1~P • [3Na] • ADP. This visible fact is

confirmed by the supplementary video. And the claim of a (notable) movement “toward the depth of cation binding cavity” may be true only for ion II in E1 · 3Na (Fig. 4e).

We improved the resolution of the cryo-EM maps of the E1·3Na and E1·3Na·ATP states to 3.2 Å and 2.9Å, respectively. In the new maps of higher resolution, site II location does almost not shift in E1·3Na and E1·3Na·ATP. After consideration, we removed this sentence about Na⁺ movement.

7.line 112: What is the meaning of “ADP has a smaller pocket by the tilt with N and P domain …” Does ADP ‘require a smaller pocket’ or ‘induces a reduction of the pocket’ ? Is the tilt of the N and P domain cause (as this sentence may claim) or effect?

We thank this reviewer for pointing this out. After consideration, we removed this sentence and explained the mechanism of gate closure by P domain bending in “*Cytoplasmic gate closure coupled with ATP hydrolysis*” section.

8.lines 210-214: The E1 · 3Na · ATP and E1 · 3Na structures DO NOT reveal a cytoplasmic gating mechanism correlated with the ATP-dependent Na⁺-binding site remodeling. If anything, their comparison with the E1~P · [3Na] · ADP structure does.

We thank this reviewer for this point. After consideration, we removed this sentence.

Supplementary Materials

9.line 29: How many/which peak fractions were collected?

We add this information in “**Materials and methods**” part of supplementary material and add a sentence “*The peak fractions (14.5-15 ml) were collected and concentrated for EM analysis.*” We also add it in figure comments of supplementary Fig. 2a “*The peak fractions (14.5-15 ml) were collected and concentrated for EM analysis.*”

10.line 94, Supplementary Fig. 2: Correction necessary, panel “f” should be “d” , and panel “i” should be “e” .

Point taken. We have corrected it.

Reviewer #3 (Remarks to the Author):

The paper by Guo et al., entitled “Cryo-EM structures of the human sodium-potassium pump revealing the gating mechanism on the cytoplasmic side.”

Describe the cryo-EM structure of the human Na, K ATPase expressed recombinantly in HEK cells and determined structural in the E1 state with three sodium ions bound in both the presence of an ATP analogue and without. The authors also present the cryoEM structure of the E2 state bound with two potassium ions. The study represents a milestone on several levels. It is the first cryo-EM structure of recombinantly produced Na, K ATPase. It will likely pave the way for understanding the several disease mutants known for this important enzyme and perhaps describing how many of the known drugs affect the Na, K ATPase, which likely will not be possible for crystallography, i. e the palytoxin bound structure.

The paper is exceptionally well written and in easy term describe how the sodium and potassium ions are bound, and confirm the position with the known crystal structures and thus will likely become the method of choice for future structural endeavours for members of the P-type ATPase family.

1. What seems to be missing is a description of the lipids, in particular the cholesterol site present in the Na, K -ATPase. At this resolution, it should be visible, and what of other lipids bound in the transmembrane region? As this paper describes a complete description of Na, K ATPase from a recombinantly expressed protein, a description of the lipid environment should be included.

We thank this reviewer for pointing this out. We have compared the lipids' locations of three our structures and other NKA structures. We add a new sentence “*Some lipid-like densities were found near site A and site C16, for which the cholesterol analogue CHS and phosphatidic acid 3PH was manually built according to the purification conditions (Supplementary Fig. 5)..*” in “Structural determination of hNKA in three conformations” part.

2. The title is overselling the paper. Only structural data is presented for a putative gating mechanism and would require functional data to be included. The title should be reduced to Cryo-EM structures of recombinant human sodium-potassium pumps determined in two different states.

Point taken. We changed the title to “Cryo-EM structures of recombinant human sodium-potassium pumps determined in three different states.” accordingly, which we agree is better.

REVIEWER COMMENTS

Reviewer #1 (Remarks to the Author):

The manuscript has been improved, but in several ways it does not solve the previous problems. Now the rotamer conformation of E334 is described in the text, EM densities in the corresponding figure (Fig S7) are hardly visible at all, either on the screen or in print. The authors need to revise the figure so that mesh is clearly visible. Figure R1 is relatively easy to see, but it is not more visible due to the increased resolution, and this density does not support either open or close of E334, it should be said that it is disordered. This reviewer strongly recommends that the EM density, at least around E334 site II Na (Figure R1), must be included in the main figure to show the reliability of pdb models, as these are directly related to the main conclusion. Experimental data indicate that the cytoplasmic gate appeared to be open as a consequence of E334 disorder, not due to E334 having an open rotamer conformation. These two things must be clearly discriminated in the presentation.

Regarding L134, it does not matter whether the author prefers the model or not. There should be a reason for building and adopting that model. Authors should describe logical reason for their model building. Absence of backing support by Phe and Leu in NaE1 and NaE1ATP, in contrast to E1P-ADP, allows E334 to take an open conformation? Is the environment in which the carboxyl group of E334 are present sufficiently hydrophilic?

The intent of my previous comment about HOLE analysis was to help present the connection of site III to the cytoplasmic solution toward M1-2. However, the results in FigS8 appear to be quite different. This is misleading, and I recommend removing this figure. If authors want to keep this figure, at least the conclusions from this analysis should be stated in the text: the pore radius of 1Å is too narrow for Na⁺ to pass through. However, by showing the tunnel, the reader can get the impression that Na⁺ may reach Site III through M4-M5.

There is no interpretation regarding calculated valence values, which is far from the ideal value of 1.0. These values should be mentioned and discussed in the text.

Minor but important points

L86: 2856 atoms? NKA contains approx. 1000 amino acids for alpha-subunit. This number must be confirmed.

L115: perhaps, among three Na?

L118-119: What does the displacement of Na-coordinating residues mean? First of all, the criterion for coordinating oxygen is not clearly defined in the manuscript: is it within 3.5Å for hydrogen bond? 4Å for salt bridge? This is also applicable for Fig.2 dotted lines. Is the difference in the coordination of site III Na between NaE1 and NaE1ATP coupled with the conformational change that occurs at the cytoplasmic side? Is there any structural reason why site III coordination is different in these two states? I am asking this because D933 is considered to be a key residue for site III Na coordination (many papers have been published about this residue).

L190: 923 atoms?

L205: ²¹⁹

L220-: The strategy of using recombinant proteins instead of natural source has already been reported in the crystallization of SERCA (Toyoshima et al., 2013 Nature) and H,K-ATPase (Abe et al., 2018, Nature), as well as in cryo-EM of H,K-ATPase (Abe et al., 2021, Nat Commun). These papers should be mentioned here.

L242: Since AMPPCP induces the identical conformation as AIF-ADP does, ATP hydrolysis does not a determinant of M1 sliding. It appeared to be triggered by the P-domain bending, which is likely induced by the interaction between gamma phosphate and catalytic Asp (and taking upright conformation, ATPgammaS does not induce this). This should be corrected.

Figure 2:

EM density for E334 must be included.

Panel b-e are too small. More close-up view (you can crop only cation-binding site. M3, M8, M10 do not necessary) is helpful to follow cation coordination.

As mentioned above, indicate the definition of dotted lines that connect Na and oxygens.

Fig3:

Panel e, human SERCA? Maybe rabbit

Fig4:

L373: no residues are displayed in the figure

L374: it is difficult to understand what does red dashed line indicate.

Fig5:

Cytoplasmic K⁺-binding is not related to the conclusion, and I don't understand why this K⁺ is highlighted with a circle. If authors want to preserve it, put a circle on the middle panel as well.

Reviewer #2 (Remarks to the Author):

The authors responded to this reviewer's comments and suggestions in appropriately.

Rereading the manuscript I realized, however, a minor inconsistency in Supplementary Figure 16 which should be considered by the authors. The leftmost schematic model is described as "E2/E1 transition", which is a process, while the scheme represents a state, namely the state immediately after the claimed transition. According to the Post-Albers scheme, in the E2 state the gate has to be closed, it opens only after the transition to E1, in the scheme the gate is mentioned to be open. Maybe, the presentation can be reconciled by an amendment to the title of the scheme: "Upon the E2/E1 transition".

REVIEWER COMMENTS

We appreciate the thoughtful and constructive comments from all reviewers and respond to each point here. The major changes are also highlighted in the revised manuscript.

Reviewer #1 (Remarks to the Author):

1. The manuscript has been improved, but in several ways it does not solve the previous problems. Now the rotamer conformation of E334 is described in the text, EM densities in the corresponding figure (Fig S7) are hardly visible at all, either on the screen or in print. The authors need to revise the figure so that mesh is clearly visible. Figure R1 is relatively easy to see, but it is not more visible due to the increased resolution, and this density does not support either open or close of E334, it should be said that it is disordered. This reviewer strongly recommends that the EM density, at least around E334 site II Na (Figure R1), must be included in the main figure to show the reliability of pdb models, as these are directly related to the main conclusion. Experimental data indicate that the cytoplasmic gate appeared to be open as a consequence of E334 disorder, not due to E334 having an open rotamer conformation. These two things must be clearly discriminated in the presentation.

We would like to thank this reviewer for the helpful comments. We earnestly accept your considerations about the conformation of E334 side chain. Firstly, we have added the EM density of M1e and M4e around E334 in the main Figure 2 to help reader better judge these data. Secondly, we have improved the Figure S7 with a clearer mesh representing the cryo EM densities around three Na⁺ binding sites. Lastly, the article has been seriously revised to describe the unstable E334 side chain conformation in line 142-145 “*Although previous studies have observed that Glu334 is a part of site II for cation binding in a close conformation¹⁸, we could not identify a stable conformation for the side chain of Glu334 due to weak density (Supplementary Fig. 4d, 7a). We speculate the side chain of Glu334 is unstable for the formation of Na⁺-coordination.*” and in line 147 and 149 “*Therefore, we speculate a gating residue Glu334 on M4, which has an unstable side chain, allows Na⁺ entrance before the M1 sliding gate closure.*”.

2. Regarding L104, it does not matter whether the author prefers the model or not. There should be a reason for building and adopting that model. Authors should describe logical reason for their model building. Absence of backing support by Phe

and Leu in NaE1 and NaE1ATP, in contrast to E1P-ADP, allows E334 to take an open conformation? Is the environment in which the carboxyl group of E334 are present sufficiently hydrophilic?

We sincerely thank the reviewer for highlighting this point and agree with his view. We add this point in the revised manuscript in line 149-153 “*Structural comparison between different E1 states shows that the Phe100 and Leu104 residues of hNKA (corresponding to residues Phe93 and Leu97 in pig NKA) are away from Glu334 in the E1·3Na·ATP state, allowing Glu334 to located in the open mouth of the entrance pathway of Na⁺ (Supplementary Fig. 8a-d).*”. We think the side chain carboxyl group of E334 is in a hydrophilic environment in the E1·3Na·ATP state.

3.The intent of my previous comment about HOLE analysis was to help present the connection of site III to the cytoplasmic solution toward M1-2. However, the results in FigS8 appear to be quite different. This is misleading, and I recommend removing this figure. If authors want to keep this figure, at least the conclusions from this analysis should be stated in the text: the pore radius of 1A is too narrow for Na⁺ to pass through. However, by showing the tunnel, the reader can get the impression that Na⁺ may reach Site III through M4-M5.

We accept this point and remove the HOLE figure from the Supplementary figure.

4.There is no interpretation regarding calculated valence values, which is far from the ideal value of 1.0. These values should be mentioned and discussed in the text.

We thank this reviewer’s suggestion. We have discussed the Na⁺ valence in revised manuscript line 128-134 “*We calculated partial valence¹⁹ for Na⁺ ions at the sites I, II, and III in the E1·3Na state, which is 0.493, 0.397, and 0.653, respectively (Supplementary Table 2), indicating site III is a reasonable coordination site for a sodium ion. The relative lower valence values for site I, II (less than to 0.520) imply that sodium ion does not bind at these sites in a stable way. Whereas in the E1·3Na·ATP state, the partial valence for the sites I, II, and III is 0.748, 0.313, and 0.717, respectively (Supplementary Table 3), increased at site I and III and decreased at site II.*”. The criterion of 0.5 for a stable valence is reported in the reference “Sauer D B, Song J, Wang B, et al. Structure and Inhibition Mechanism of the Human Citrate Transporter Nact[J]. Nature, 2021, 591(7848):157-161.”.

Minor but important points

5.L86: 2856 atoms? NKA contains approx. 1000 amino acids for alpha-subunit. This number must be confirmed.

We have corrected the atom number in revised manuscript in line 88-89 “*The structures of the E1·3Na and E1·3Na·ATP states are very similar to each other, with a root mean square deviation (RMSD) of 1.388 Å of 976 Ca atoms from α subunit, ...*”.

6.L115: perhaps, among three Na?

We thank this reviewer’s suggestion and changed the text to “*Site III is located between M5, M6 and M8 and the most restricted among three Na⁺ binding sites (Fig. 2d-e, Supplementary Fig. 8a-d).*” in line 117–118.

7.L118-119: What does the displacement of Na-coordinating residues mean? First of all, the criterion for coordinating oxygen is not clearly defined in the manuscript: is it within 3.5Å for hydrogen bond? 4Å for salt bridge? This is also applicable for Fig. 2 dotted lines. Is the difference in the coordination of site III Na between NaE1 and NaE1ATP coupled with the conformational change that occurs at the cytoplasmic side? Is there any structural reason why site III coordination is different in these two states? I am asking this because D933 is considered to be a key residue for site III Na coordination (many papers have been published about this residue).

The displacement of Na-coordinating residues means that there is conformational change in Na⁺ ion binding sites. The criterion for the Na⁺-coordinating residues is 4 Å in this paper. We added this criterion in Fig. 2 legend for more clear description of the dotted lines: “*The key residues coordinated with Na⁺ within 4 Å are shown in stick representation and indicated by dotted lines.*”. There are only minor structural differences between the E1·3Na and E1·3Na·ATP states, so it’s hard to determine whether the subtle changes of site III location are coupled with the conformational change that occurs at the cytoplasmic side when ATP binding or not. To make the statement of the Na⁺-coordinating residues more clear and avoid confusion, we have adjusted the sentence in manuscript “*The Thr781, Thr814 and Gln930 residues are involved in Na⁺ coordination at site III in the E1·3Na state, whereas the Asp933*

residue is close to and coordinated with Na⁺ in the E1·3Na·ATP state.” in line 119-122.

8.L190: 923 atoms?

We have corrected the atom number in revised manuscript in line 203-206
“*Compared to the E2·[2K]·Pi state (a preceding state in the Post-Albers cycle)8, the two structures are very similar, with an RMSD of 1.083 Å between 994 Ca atom pairs of a subunit (Supplementary Fig. 10).”*

9.L205: ²¹⁹

It is changed as suggested.

10.L220-: The strategy of using recombinant proteins instead of natural source has already been reported in the crystallization of SERCA (Toyoshima et al., 2013 Nature) and H,K-ATPase (Abe et al., 2018, Nature), as well as in cryo-EM of H,K-ATPase (Abe et al., 2021, Nat Commun). These papers should be mentioned here.

We thank this reviewer for pointing this out. Following the detailed instruction, we carefully read these papers and have added these references in our revised manuscript in line 234–235 “... *although recombinant proteins were already used to study the other P2-type members (SERCA31 and H⁺,K⁺-ATPase32).* ”.

11.L242: Since AMPPCP induces the identical conformation as AIF-ADP does, ATP hydrolysis does not a determinant of M1 sliding. It appeared to be triggered by the P-domain bending, which is likely induced by the interaction between gamma phosphate and catalytic Asp (and taking upright conformation, ATPgammaS does not induce this). This should be corrected.

We thank this reviewer for pointing this out and corrected it in the revised manuscript “**Discussion**” part in line 252–266 “ *The second gating stage at the cytoplasmic side is the gate closure during the transition from the E1·3Na·ATP to E1~P·[3Na]·ADP state (Fig. 4 and Supplementary Fig. 8). The M1 sliding door is elevated to close the cytoplasmic gate, accompanying the dramatically conformational changes with closed headpiece and an unbent P domain that was proposed in a recent publication about SERCA²⁵. The bending of the P domain is required for making phosphoryl transfer possible. Intriguingly, our data imply that ATP γ S binding mode in hNKA is different from ACP in SERCA E1·2Ca·ATP state^{18,33}, but is similar to ACP in SERCA E2·ACP state²⁵ (Fig. 3 and Supplementary Fig. 9a). ACP in E1·2Ca·ATP is able to induce PC domain in a similar unbent conformation as AlF₄- and ADP do in NKA13. We found that Asp376 side chain has an upright orientation, which is likely induced by the interaction between γ phosphate of ATP and catalytic Asp376, is very important for the gate closure mechanism by causing the PC domain bending releasing motion (Fig. 4b-c) by comparing ATP γ S, ACP, and ADP binding pockets. This may be the reason for M1 sliding door closed coupling NKA phosphorylation to Na⁺ binding site occlusion.”.*

12. Figure 2: EM density for E334 must be included.

Panel b-e are too small. More close-up view (you can crop only cation-binding site. M3, M8, M10 do not necessary) is helpful to follow cation coordination. As mentioned above, indicate the definition of dotted lines that connect Na and oxygens.

It is changed as suggested.

13. Fig3: Panel e, human SERCA? Maybe rabbit

We thank this reviewer for pointing this out and we check it in the reference “Kabashima Y, Ogawa H, Nakajima R, et al. What Atp Binding Does to the Ca(2+) Pump and How Nonproductive Phosphoryl Transfer Is Prevented in the Absence of Ca(2)[J]. Proc Natl Acad Sci U S A, 2020, 117(31):18448-18458.”. In methods section of this paper, we found “*pFN21K vector (Promega) was used for fusing HaloTag at the N terminus of human SERCA2a. Coding sequence of HaloTag-fused SERCA2a including IRES and AcGFP was subcloned into pShuttle-CMV vector (Stratagene).*”

14.Fig4:L373: no residues are displayed in the figure

L374: it is difficult to understand what does red dashed line indicate.

We delete the words “*with the key residues around them*” and changed the sentence as “*The ATP and ADP binding pockets between the N and P domains are shown in **b***”.

The dashed lines indicated the bending conformational change of the P domain. To avoid confusion, we modified the figure legend as “*The P_N and P_C half domains have a bent conformation in the $E1 \cdot 3Na \cdot ATP$ state (indicated by the red dashed line) or an unbent confirmation in the $E1\sim P \cdot [3Na] \cdot ADP$ state (indicated by the black dashed line), respectively.*”.

15.Fig5:Cytoplasmic K^+ -binding is not related to the conclusion, and I don't understand why this K^+ is highlighted with a circle. If authors want to preserve it, put a circle on the middle panel as well.

We thank this reviewer's suggestion and the red circle has been removed.

Reviewer #2 (Remarks to the Author):

The authors responded to this reviewer's comments and suggestions in appropriately. We thank this reviewer's support.

1.Rereading the manuscript I realized, however, a minor inconsistency in Supplementary Figure 16 which should be considered by the authors. The leftmost schematic model is described as “E2/E1 transition”, which is a process, while the scheme represents a state, namely the state immediately after the claimed transition. According to the Post-Albers scheme, in the E2 state the gate has to be closed, it opens only after the transition to E1, in the scheme the gate is mentioned to be open. Maybe, the presentation can be reconciled by an amendment to the title of the scheme: “Upon the E2/E1 transition”.

We thank this reviewer for pointing this out. The previous Supplementary Figure 16 is renumbered as Supplementary Figure 15 in present manuscript. The figure legend has been revised to “The first stage is gate open towards intracellular side after the transition from E2·[2K] to E1 state.”. We also changed the scheme title from “E2/E1 transition” to “Upon the E2/E1 transition” in the Supplementary Figure 15.

REVIEWERS' COMMENTS

Reviewer #1 (Remarks to the Author):

Now the authors reply to reviewer's concerns appropriately.

I hope authors could improve Fig S7 more visible. White mesh is still difficult to see, and I cannot find V329 and D933 residues in the figure. Please consider to make enough visible one when publish.

## REVIEWER COMMENTS

We appreciate the thoughtful and constructive comments from all reviewers and respond to each point here:

**Reviewer #1 (Remarks to the Author):**

Now the authors reply to reviewer's concerns appropriately.
I hope authors could improve Fig S7 more visible. White mesh is still difficult to see, and I cannot find V329 and D933 residues in the figure. Please consider to make enough visible one when publish.

We thank this reviewer's suggestion. We have carefully improved the Supplementary Fig7 (as follow) by ChimeraX software in supplementary information file, now the cryo-EM densities were shown clearly.